# Cytosolic lipid droplets as engineered organelles for production and accumulation of terpenoid biomaterials in leaves

Radin Sadre [1,2,3], Peiyen Kuo [2], Jiaxing Chen [2], Yang Yang[2,3,4], Aparajita Banerjee [2,3], Christoph Benning [2,3,4,5] & Bjoern Hamberger [2,3]

Cytosolic lipid droplets are endoplasmic reticulum-derived organelles typically found in seeds as reservoirs for physiological energy and carbon to fuel germination. Here, we report synthetic biology approaches to co-produce high-value sesqui- or diterpenoids together with lipid droplets in plant leaves. The formation of cytosolic lipid droplets is enhanced in the transient *Nicotiana benthamiana* system through ectopic production of WRINKLED1, a key regulator of plastid fatty acid biosynthesis, and a microalgal lipid droplet surface protein. Engineering of the pathways providing the universal C5-building blocks for terpenoids and installation of terpenoid biosynthetic pathways through direction of the enzymes to native and non-native compartments boost the production of target terpenoids. We show that anchoring of distinct biosynthetic steps onto the surface of lipid droplets leads to efficient production of terpenoid scaffolds and functionalized terpenoids. The co-produced lipid droplets "trap" the terpenoids in the cells.

[1] Department of Horticulture, Michigan State University, East Lansing, MI 48824, USA. [2] Department of Biochemistry and Molecular Biology, Michigan State University, East Lansing, MI 48824, USA. [3] Great Lakes Bioenergy Research Center, Michigan State University, East Lansing, MI 48824, USA. [4] MSU-DOE Plant Research Laboratory, Michigan State University, East Lansing, MI 48824, USA. [5] Department of Plant Biology, Michigan State University, East Lansing, MI 48824, USA. Correspondence and requests for materials should be addressed to R.S. (email: sadre@msu.edu) or to B.H. (email: hamberge@msu.edu)

Cytosolic lipid droplets are dynamic organelles typically found in seeds as reservoirs for physiological energy and carbon in form of triacylglycerol (oil) to fuel germination. They are derived from the endoplasmic reticulum (ER) where newly synthesized triacylglycerol accumulates in lens-like structures between the leaflets of the membrane bilayer[1]. After growing in size, the structures bud off from the outer membrane of the ER. A mature lipid droplet is composed of a hydrophobic core of triacylglycerol surrounded by a phospholipid monolayer and coated with lipid droplet associated proteins involved in the biogenesis and function of the organelle. In seeds, oleosin proteins coat and stabilize small lipid droplets preventing coalescence. These proteins contain surface—oriented amphipathic N- and C-termini essential to efficiently emulsify lipids and a conserved hydrophobic central domain anchoring the oleosins onto the surface of lipid droplets[2].

Due to the potential economical relevance of plant lipids as renewable resource for the production of high-density biofuels, strategies have been established to enhance the accumulation of triacylglycerol in vegetative tissues of high-biomass yielding crops[3–5]. A primary target for increasing lipid production has been engineering the expression of WRINKLED1. The gene encodes a member of the AP2/EREBP family of transcription factors and master regulator of fatty acid biosynthesis in seeds[6,7]. Its ectopic production in vegetative tissues promotes fatty acid synthesis in the plastids and, indirectly, triacylglycerol accumulation in lipid droplets[4,8–10]. Yields of triacylglycerol were further increased by removal of an intrinsically disordered region of Arabidopsis thaliana WRINKLED1 (AtWRI1$^{1–397}$) increasing the protein's stability[11] and through engineered co-production of WRINKLED1 with ectopic lipid biosynthesis enzymes and a plant lipid droplet associated protein[3,9].

Plant-derived terpenoids have a wide range of industrial uses such as specialty fuels, agrochemicals, fragrances, nutraceuticals, and pharmaceuticals. The limited economic sustainability of formal (petro-) chemical synthesis, or extraction and purification from the native plant source has motivated biotechnological approaches to produce industrially relevant terpenoids[12–15]. Plants accumulating high levels of terpenoids have evolved specialized anatomical features for their biosynthesis and storage including laticifer cells, resin ducts or cavities, and glandular trichomes[16]. The recently reported accumulation of terpenoids together with neutral lipids in lipid droplets in the outer root cork cells of Plectranthus barbatus (synonym Coleus forskohlii), was suggested as a mechanism to enrich and sequester the bioactive defense compounds intracellularly[17]. The co-occurrence of lipids and terpenoids invites opportunities for biotechnology to engineer the high-yield production and storage of terpenoids in vegetative tissues of lipid droplet-accumulating biomass crops. In plants, the C5-building blocks of terpenoids, dimethylallyl diphosphate (DMADP) and isopentenyl diphosphate (IDP), are synthesized by two compartmentalized pathways. Both precursor pathways represent interesting targets for biological engineering[13,18–20]. The mevalonate (MEV) pathway converts acetyl-CoA by enzyme activities located in the cytosol, ER and peroxisomes, providing precursors for a wide range of terpenoids with diverse functions such as in growth and development, defense and protein prenylation. The enzyme 3-hydroxy-3-methylglutaryl-CoA reductase (HMGR) catalyzes the rate-limiting step in the MEV pathway and engineering and production of the catalytic domain of HMGR by N-terminal truncation improved the flux of precursors into terpenoid biosynthesis[15,16,21]. Only recently, it was shown that flux through the MEV pathway is, in part, also limited by phosphomevalonate kinase (PMK) which acts downstream of HMGR[22]. The same study provided evidence that isopentenyl diphosphate kinases

and hydrolases of the Nudix superfamily are involved in determining the ratio of IDP to isopentenyl phosphate and possibly, the ratios of DMADP to dimethylallyl phosphate and farnesyl diphosphate (FDP) to farnesyl phosphate[22]. Isopentenyl diphosphate kinases reactivate isopentenyl phosphate (IP) through phosphorylation to IDP whereas hydrolases of the Nudix superfamily catalyze the dephosphorylation of IDP.

In the plastid, the 2-C-methyl-D-erythritol 4-phosphate (MEP) pathway uses pyruvate and D-glyceraldehyde 3-phosphate to provide precursors for the biosynthesis of terpenoids related to development, photosynthesis, and defense against biotic and abiotic stresses. The enzyme 1-deoxy-D-xylulose 5-phosphate synthase (DXS) is rate-limiting in the MEP pathway and its constitutive overproduction enhanced terpenoid production in some, but not all plant species tested[14,23,24]. Head-to-tail condensation of DMADP and IDP affords linear isoprenyl diphosphates, such as FDP (C15) or geranylgeranyl diphosphate (GGDP, C20) catalyzed by farnesyl diphosphate synthase (FDPS) and geranylgeranyl diphosphate synthase (GGDPS), respectively. In Nicotiana benthamiana, both DXS and GGDPS were required to enhance terpenoid synthesis[24]. Cytosolic sesquiterpene synthases and plastidial diterpene synthases convert FDPS and GGDPS, respectively, into typically cyclic terpenoid scaffolds, contributing to the enormous structural diversity among terpenoids in the plant kingdom. Such terpenoid scaffolds often undergo further stereo- and regio-selective functionalization catalyzed by ER membrane-bound mono-oxygenases, cytochromes P450 (CYPs), requiring electrons provided by co-localized NADPH-dependent cytochrome P450 reductases (CPRs).

Despite inherent advantages, such as (native) compartments and availability of reduction equivalents in form of NADPH, terpenoid biotechnology in photosynthetic tissues has remained challenging, as the engineered pathways have to compete for precursors with highly networked native pathways (and their associated regulatory mechanisms). In the present study, we establish methods towards the high-yield production of target terpenoids in leaves co-engineered for triacylglycerol accumulation in lipid droplets in the transient N. benthamiana system. Enhanced precursor flux and targeting of terpenoid synthesis enzymes to native and non-native compartments increase terpenoid production. We demonstrate that the lipid droplets sequester produced terpenoids and are suitable organelles to anchor terpenoid biosynthesis steps. By fusing terpenoid enzymes to a microalgal lipid droplet surface protein, terpenoid production is successfully re-targeted to lipid droplets. Our findings will have implications for future generation of stably transformed biomass crops efficiently producing industrially relevant terpenoids in photosynthetic tissues.

## Results

**Engineered triacylglycerol accumulation.** NoLDSP, a lipid droplet surface protein from the microalga Nannochloropsis oceanica, has functions partially analogous to plant oleosins[25]. Similar to oleosins, NoLDSP possesses a hydrophobic central region that likely mediates the anchoring on lipid droplets. To assess the impact of NoLDSP on AtWRI1$^{1–397}$-initiated triacylglycerol accumulation, we infiltrated leaves of N. benthamiana with Agrobacterium tumefaciens suspensions for transient production of AtWRI1$^{1–397}$ alone or in combination with NoLDSP (AtWRI1$^{1–397}$ + NoLDSP). In leaves producing AtWRI1$^{1–397}$ or AtWRI1$^{1–397}$ + NoLDSP, the triacylglycerol level was ~3-fold and 12-fold higher, respectively, than in control leaves without AtWRI1$^{1–397}$ (Fig. 1a). The results clearly demonstrated that the microalgal NoLDSP had no negative impact on triacylglycerol

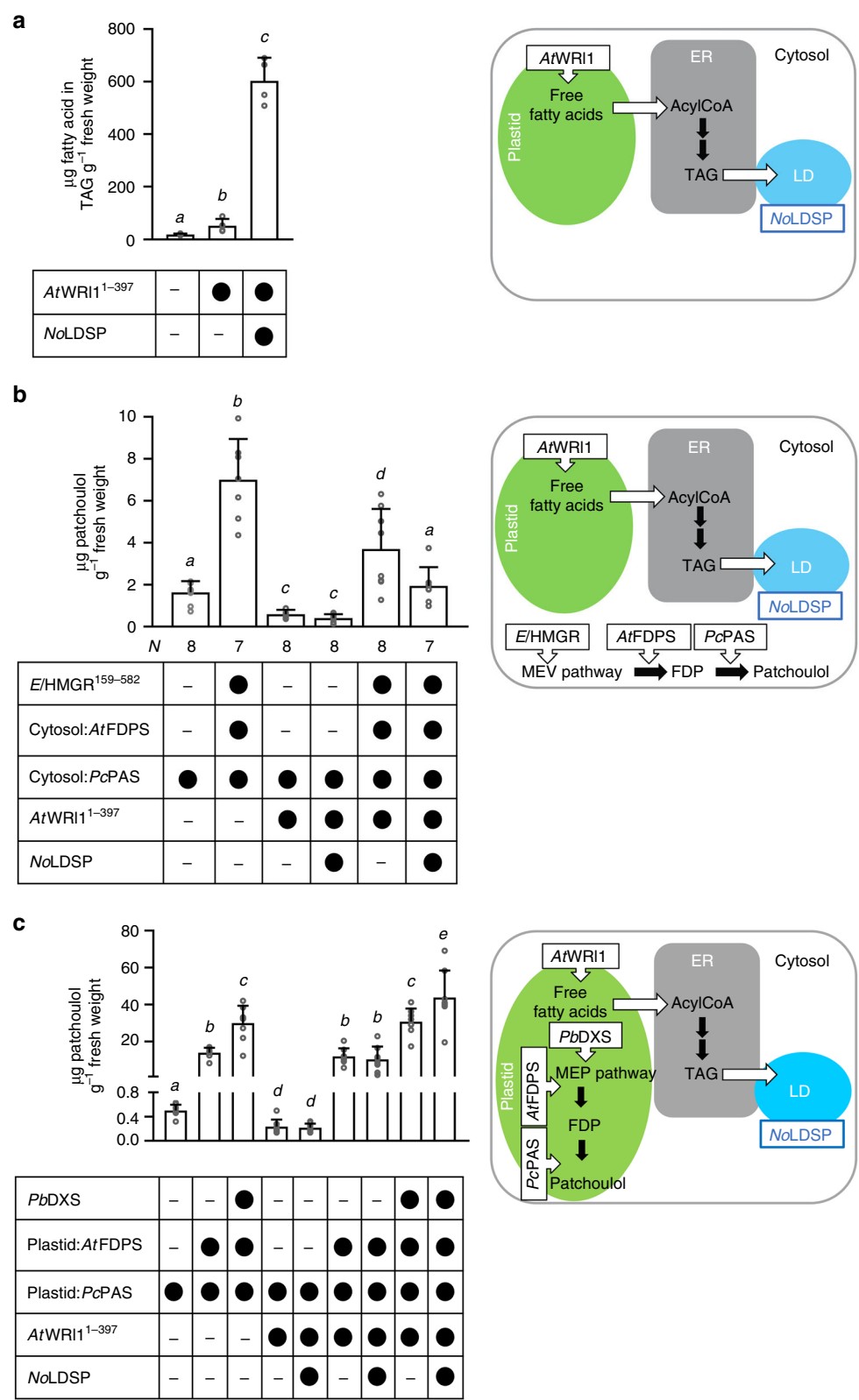

production and enhanced the accumulation of lipid droplets in infiltrated *N. benthamiana* leaves.

**Sesquiterpenoid production in the cytosol and plastids.** We then tested different engineering strategies for the production of sesquiterpenoids using patchoulol as a model compound. Like

many other sesquiterpenoids, patchoulol is volatile and its engineered production in transgenic lines of *N. tabacum* resulted in significant losses from volatile emission[15]. In our study, losses by atmospheric terpenoid emission were not recorded as the engineering strategies were designed to sequester target terpenoids in lipid droplets in the plant biomass. Transient production of cytosolic *Pogostemon cablin* patchoulol synthase (cytosol:*Pc*PAS)

**Fig. 1** Engineered patchoulol production in *N. benthamiana* leaves. Triacylglycerol (TAG) accumulation was initiated through expression of *WRINKLED1* (*At*WRI1$^{1-397}$) and further enhanced through co-expression of *No*LDSP (**a**). Patchoulol production was engineered in the cytosol (**b**) and in the plastid (**c**) in the absence and presence of *At*WRI1$^{1-397}$ and *No*LDSP. To enhance FDP availability for patchoulol production, *El*HMGR$^{159-582}$ (cytosol), *Pb*DXS (plastid) and *At*FDPS (cytosol, plastid) were included in transient assays. The different construct combinations are indicated below each bar (black circle, was included; minus, was not included) and in the scheme next to each graph. Data were analyzed by Shapiro–Wilk, Welch–ANOVA (**a** $P < 0.0006$; **b** $P < 0.0001$; **c** $P < 0.0001$) and Brown–Forsythe ANOVA (**a** $P < 0.0004$; **b** $P < 0.0001$; **c** $P < 0.0001$) followed by *t*-tests (unpaired, two-tailed, Welch correction). Data are presented as individual biological replicates and bars representing average levels with SD (**a** $N = 4$; **b** $N$ indicated below each bar; **c** $N = 8$). This experiment was replicated twice. Statistically significant differences are indicated by *a–e* based on *t*-tests ($P < 0.05$). Source Data are provided as a Source Data file. LD, lipid droplet

led to formation of a single low-level product, patchoulol, which was not detected in wild-type control plants (Fig. 1b). To enhance the precursor availability for sesquiterpenoid synthesis, a feedback-insensitive form of *Euphorbia lathyris* HMGR (*El*HMGR$^{159-582}$) and *A. thaliana* FDPS (cytosol:*At*FDPS) were included in the transient assays. *E. lathyris* accumulates high levels of triterpenoids and their esters[26], suggesting that its HMGR could be a robust enzyme for sesquiterpenoid production in *N. benthamiana*. The selection of the *A. thaliana* FDPS was based on its relatively high thermal stability[27]. The patchoulol content in *N. benthamiana* leaves producing *El*HMGR$^{159-582}$ + cytosol:*At*FDPS + cytosol:*Pc*PAS was ~5-fold higher than in leaves with cytosol:*Pc*PAS which is consistent with enhanced precursor flux. Co-engineering of patchoulol and triacylglycerol synthesis impaired cytosolic terpenoid accumulation, independent of whether precursor availability was increased or not (Fig. 1b).

A previous study demonstrated that re-direction of *Pc*PAS and avian FDPS to the plastid increased the (retained) patchoulol level in leaves of stable transgenic *N. tabacum* lines up to ~30 µg patchoulol g$^{-1}$ fresh weight[15]. We modified this approach to further examine engineering strategies for the co-production of patchoulol and lipid droplets in *N. benthamiana* leaves. Targeting of patchoulol synthase to the plastids (plastid:*Pc*PAS) led to accumulation of approximately 0.5 µg patchoulol g$^{-1}$ fresh weight (Fig. 1c). To increase the precursor flux in the plastids, *P. barbatus* DXS (*Pb*DXS) and plastid-targeted *At*FDPS (plastid:*At*FDPS) were combined with plastid:*Pc*PAS in the assays. This strategy resulted in a 60-fold increase in the level of patchoulol (Fig. 1c). Synthetic lipid droplet accumulation impaired patchoulol production in leaves in the absence of *Pb*DXS and plastid:*At*FDPS, when precursor synthesis was not co-engineered (Fig. 1c). The negative impact of lipid droplet production on patchoulol synthesis was rescued when plastid:*At*FDPS or *Pb*DXS + plastid:*At*FDPS were included in the assay. Leaves transiently producing *Pb*DXS + plastid:*At*FDPS + plastid:*Pc*PAS + *At*WRI1$^{1-397}$ + *No*LDSP yielded the highest patchoulol level retained in leaves (up to ~45 µg patchoulol g$^{-1}$ fresh weight), an average 90-fold and 1.5-fold higher compared to leaves producing plastid:*Pc*PAS and *Pb*DXS + plastid:*At*FDPS + plastid:*Pc*PAS, respectively.

**Diterpenoid scaffold production in plastids and cytosol.** Strategies for diterpenoid production in the *N. benthamiana* system were examined using the *Abies grandis* abietadiene synthase (*Ag*ABS) as diterpene synthase[28,29]. The bifunctional enzyme has class II and class I terpene synthase activity and catalyzes both the bicyclization of GGDP to a (+)-copalyl diphosphate intermediate and the subsequent secondary cyclization and further rearrangement. Transient production of the native plastidial *A. grandis* abietadiene synthase (plastid:*Ag*ABS) resulted in the accumulation of abietadiene (abieta-7,13-diene), levopimaradiene (abieta-8 (14),12-diene), neoabietadiene (abieta-8(14),13(15)-diene) and, as

minor product, palustradiene (abieta-8,13-diene) consistent with the previous findings[30]. These diterpenoids were not detected in wild-type control leaves of *N. benthamiana*. Sole production of plastid:*Ag*ABS yielded ~40 µg diterpenoids g$^{-1}$ fresh weight (Fig. 2a). To enhance the production of diterpenoids, plastid: *Ag*ABS was co-produced in different combinations with *Pb*DXS and a plastid GGDPS. GGDPSs are differentiated into three types (type I–III) according to their amino acid sequences around the first aspartate-rich motif. These three types differ in their mechanism of determining product chain-length[31,32]. Plant GGDPSs are type II enzymes that are regulated on gene expression, transcript, and protein level[33–35]. We hypothesized that inclusion of distantly related type I and type III GGDPSs or a cyanobacterial type II GGDPS may allow us to bypass potential regulatory steps limiting diterpenoid production in *N. benthamiana*. Six GGDPSs were selected: an archaeal GGDPS from *Sulfolobus acidocaldarius* (*Sa*GGDPS, type I), a predicted archaeal GGDPS from *Methanothermobacter thermautotrophicus* (*Mt*GGDPS, type I), a predicted cyanobacterial GGDPS from *Tolypothrix* sp. PCC 7601 (*Ts*GGDPS, type II), two predicted plant GGDPSs from *Euphorbia peplus* (*Ep*GGDPS1 and *Ep*GGDPS2, type II), and one predicted GGDPS from the fungus *Mortierella elongata* AG77 (*Me*GGDPS, type III). *Sa*GGDPS, *Mt*GGDPS, and *Me*GGDPS share only 24%, 25 and 17% amino acid identities with *Ep*GGDPS1, respectively, whereas *Ts*GGDPS and *Ep*GGDPS2 share 48 and 58% identities with *Ep*GGDPS1, respectively. For transient assays in *N. benthamiana*, the coding sequences for the bacterial and fungal GGDPSs were codon-optimized (except for *Ts*GGDPS) and modified to target the enzymes to the plastids, referred to as plastid:*Sa*GGDPS, plastid: *Mt*GGDPS, plastid:*Ts*GGDPS and plastid:*Me*GGDPS. Co-production of *Pb*DXS + plastid:*Ag*ABS or plastid:GGDPS + plastid:*Ag*ABS was insufficient to increase the diterpenoid content in *N. benthamiana* leaves more than 2-fold compared to the diterpenoid level in plastid:*Ag*ABS-producing leaves (Fig. 2a). In contrast, co-production of *Pb*DXS + GGDPS + plastid:*Ag*ABS enhanced diterpenoid production up to 6.5-fold compared to leaves producing plastid:*Ag*ABS). Significant differences in diterpenoid yields were obtained depending on which GGDPS was included, apparently unrelated to a specific type of GGDPS (Fig. 2a). The highest diterpenoid levels were determined in *N. benthamiana* leaves co-producing *Pb*DXS + plastid:*Ag*ABS with plastid:*Mt*GGDPS (type I), plastid:*Ts*GGDPS (type II), or *Ep*GGDPS2 (type II), with similar yield between these combinations (Fig. 2a).

We further evaluated diterpenoid accumulation in the presence of lipid droplets. Co-production of plastid:*Ag*ABS + *At*WRI1$^{1-397}$ had no significant impact on the diterpenoid level compared to control leaves producing plastid:*Ag*ABS, whereas in leaves producing plastid:*Ag*ABS + *At*WRI1$^{1-397}$ + *No*LDSP, the diterpenoid content was increased 2-fold (Fig. 2b). Similarly, co-production of plastid:*Mt*GGDPS + plastid:*Ag*ABS + *At*WRI1$^{1-397}$ + *No*LDSP increased the diterpenoid level 2.5-fold compared to plastid: *Mt*GGDPS + plastid:*Ag*ABS producing leaves. The results indicated

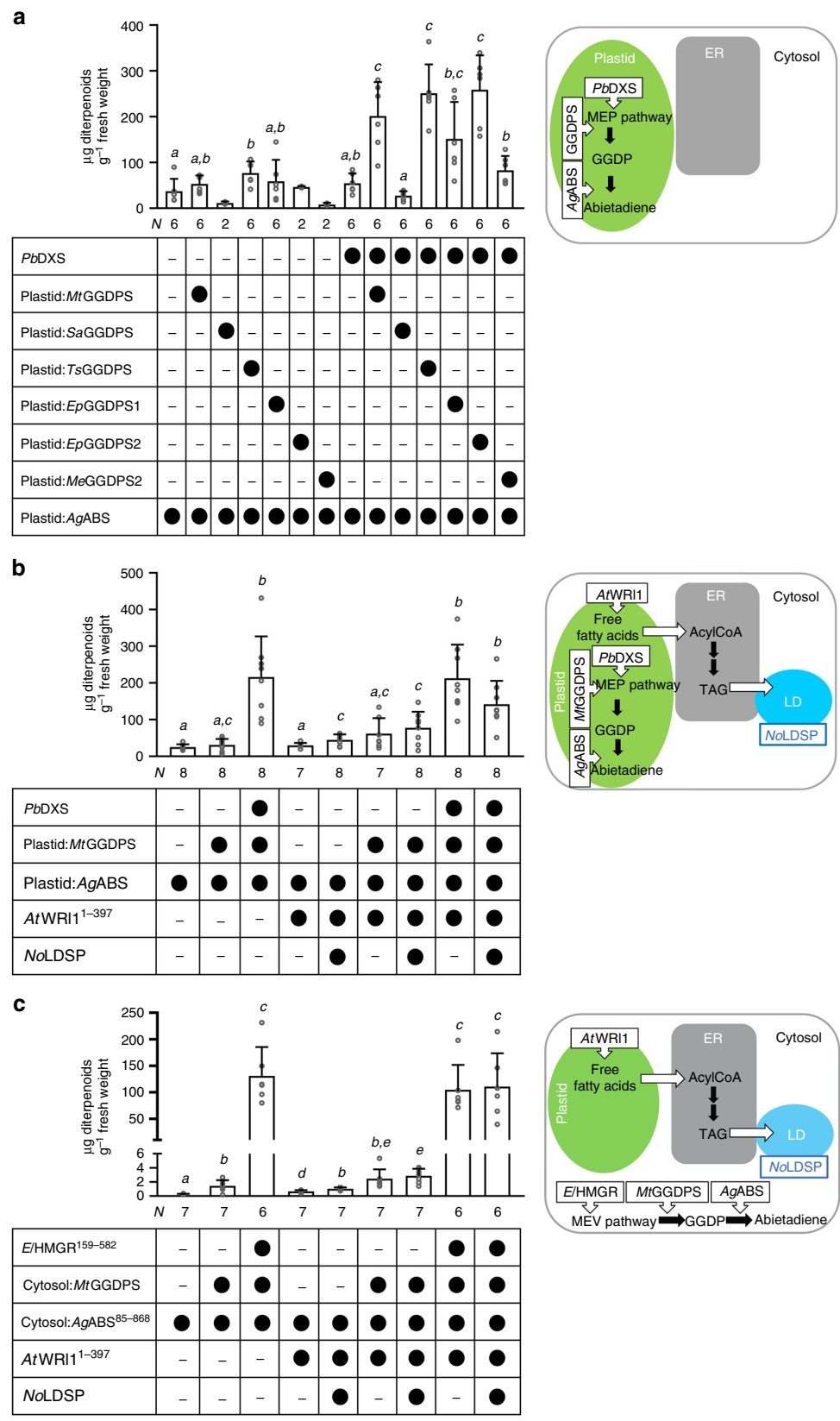

that the increased abundance of lipid droplets was beneficial for the accumulation of diterpenoid products. Sequestration of the lipophilic citerpenoids into lipid droplets may have helped to circumvent negative feedback regulatory mechanisms and served as "pull force" in diterpenoid production. In fact, isolated lipid droplet fractions from leaves producing plastid:$Ag$ABS + $At$WRI1$^{1-397}$ and plastid:$Ag$ABS + $At$WRI1$^{1-397}$ + $No$LDSP contained ~35-fold and 460-fold more diterpenoids, respectively, than control fractions from leaves with plastid:$Ag$ABS, consistent with the sequestration of diterpenoids in lipid droplets (Supplementary Fig. 1). Co-

**Fig. 2** Engineered diterpenoid production in *N. benthamiana* leaves. Production of *Ag*ABS led to accumulation of diterpenoids (abietadiene and its isomers). To enhance GGDP availability for diterpenoid production, *El*HMGR$^{159-582}$ (cytosol), *Pb*DXS (plastid), and distinct GGDPSs (cytosol or plastid) were included in transient assays. The protein combinations are indicated below each bar (black circle, was included; minus, was not included) and in the scheme next to each graph. The production of diterpenoids was engineered in the plastid (**a**, **b**) and in the cytosol (**c**) in the absence and presence of *At*WRI1$^{1-397}$ and *No*LDSP. Data were analyzed by Shapiro–Wilk, Welch–ANOVA (**a** *P* < 0.0001; **b** *P* < 0.0001; **c** *P* < 0.0001) and Brown–Forsythe ANOVA (**a** *P* < 0.0001; **b** *P* < 0.0001; **c** *P* < 0.0001) followed by *t*-tests (unpaired, two-tailed, Welch correction). Data are presented as individual biological replicates and bars representing average levels with SD (*N* indicated below each bar). This experiment was replicated twice. Statistically significant differences are indicated by *a-e* based on *t*-tests (*P* < 0.05). Source Data are provided as a Source Data file. LD, lipid droplet

production of *Pb*DXS and plastid:*Mt*GGDPS together with plastid:*Ag*ABS yielded the highest diterpenoid level (Fig. 2b) independent of whether *At*WRI1$^{1-397}$ was included for lipid droplet synthesis. In contrast, co-production of *Pb*DXS + plastid:*Mt*GGDPS + plastid:*Ag*ABS together with *At*WRI1$^{1-397}$ + *No*LDSP resulted in a significant reduction of the diterpenoid level (compared to leaves producing *Pb*DXS + plastid:*Mt*GGDPS + plastid:*Ag*ABS).

When *A. grandis* abietadiene synthase was targeted to the cytosol (cytosol:*Ag*ABS$^{85-868}$), leaves accumulated ~0.2 μg diterpenoids g$^{-1}$ fresh weight and addition of precursor pathway genes enhanced diterpenoid synthesis (Fig. 2c). Co-production of cytosol:*Ag*ABS$^{85-868}$ together with *El*HMGR$^{159-582}$ and cytosolic *M. thermautotrophicus* GGDPS (cytosol:*Mt*GGDPS) increased the diterpenoid yield more than 400-fold (relative to cytosol:*Ag*ABS$^{85-868}$ containing leaves) and, thus, close to the highest diterpenoid yield achieved with plastid engineering approaches (Fig. 2b, c). Moreover, our data indicated an enhancing effect of lipid droplet accumulation on terpenoid production when cytosol:*Ag*ABS$^{85-868}$ was co-produced with *At*WRI1$^{1-397}$ or *At*WRI1$^{1-397}$ + *No*LDSP (Fig. 2c). Under these conditions, terpenoid production was increased up to approximately 3-fold which is consistent with diterpenoids being sequestered in lipid droplets. When *El*HMGR$^{159-582}$ + cytosol:*Mt*GGDPS + cytosol:*Ag*ABS$^{85-868}$ + *At*WRI1$^{1-397}$ + *No*LDSP were co-produced, no additive effects of lipid droplet engineering on terpenoid yield were detected (relative to *El*HMGR$^{159-582}$ + cytosol:*Mt*GGDPS + cytosol:*Ag*ABS$^{85-868}$) (Fig. 2c).

**Triacylglycerol analysis of *N. benthamiana* leaves**. To examine a potential impact of terpenoid engineering on triacylglycerol yield, the established approaches for low- or high-yield terpenoid synthesis combined with lipid droplet production were further tested. Four days after infiltration, the leaves were subjected to triacylglycerol analysis. Leaves co-engineered for lipid droplet and patchoulol production in the cytosol contained ~50% less triacylglycerol than leaves producing *At*WRI1$^{1-397}$ + *No*LDSP (Fig. 3a). A significant decrease in the triacylglycerol level was also detected when leaves were engineered for cytosol-targeted high-yield production of diterpenoids (compared to leaves producing *At*WRI1$^{1-397}$ + *No*LDSP) (Fig. 3b). When lipid droplet production was combined with a plastid-targeted approach for high-yield terpenoid synthesis, no negative impact on triacylglycerol accumulation was observed compared to control plants (Fig. 3a, b).

**Targeting diterpenoid production to lipid droplets**. We next investigated whether lipid droplets in the cytosol can be used as platform to anchor biosynthetic pathways for the production of functionalized diterpenoids. The proof-of-concept experiments included modified *A. grandis* abietadiene synthase and *Picea sitchensis* cytochrome P450 (*Ps*CYP720B4), previously reported to convert abietadiene and several isomers to the corresponding diterpene resin acids[36]. To target terpenoid synthesis to the lipid droplets, *A. grandis* abietadiene synthase lacking the N-terminal

plastid targeting sequence (cytosol:*Ag*ABS$^{85-868}$) and truncated *Ps*CYP720B4 lacking the N-terminal membrane-binding domain (cytosol:*Ps*CYP720B4$^{30-483}$) were produced as C-terminal and N-terminal *No*LDSP-fusion protein, respectively. The *No*LDSP-fusion proteins are here referred to as LD:*Ag*ABS$^{85-868}$ and LD:*Ps*CYP720B4$^{30-483}$. The construction of LD:*Ag*ABS$^{85-868}$ as C-terminal *No*LDSP-fusion protein was inspired by studies reporting on functional, C-terminal tagged diterpene synthases[37,38]. To re-target *Ps*CYP720B4 to lipid droplets (LD:PsCYP720B4$^{30-483}$), the predicted N-terminal hydrophobic domain of native *Ps*CYP720B4 was replaced by *No*LDSP as a recent publication described that modifications or deletion of the membrane anchoring domain of CYP720B4 did not impair the enzyme's activity[38]. Inclusion of CPRs has been shown to be crucial to drive metabolic fluxes in CYP-mediated production of high-value target compounds in non-native hosts and synthetic compartments[39,40]. In our experiments, *Camptotheca acuminata* CPR (cytosol:*Ca*CPR$^{70-708}$) was included as *No*LDSP-fusion protein to co-localize the *Ca*CPR and *Ps*CYP720B4 activities on lipid droplets and facilitate the CYP-catalyzed production of functionalized terpenoids. As the C-terminus of CPRs is pivotal for catalytic activity and not suitable for modifications[41,42], the predicted N-terminal hydrophobic domain of native *Ca*CPR was replaced by *No*LDSP to produce the fusion protein LD:*Ca*CPR$^{70-708}$.

To determine the localization *in planta*, the *No*LDSP-fusion proteins were each produced as yellow fluorescent protein (YFP)-tagged proteins together with *At*WRI1$^{1-397}$ for lipid droplet production. The YFP-signals in infiltrated leaves were subsequently compared to the signals obtained for YFP-tagged *No*LDSP, which indicated that all three YFP-tagged *No*LDSP-fusion proteins were targeted to the surface of the lipid droplets (Fig. 4). It is noteworthy that production of the YFP-tagged *No*LDSP and *No*LDSP-fusion proteins promoted clustering of small lipid droplets in planta and in isolated lipid droplet fractions, consistent with a previous report on ectopic production of *A. thaliana* OLEOSIN1 fused to green fluorescent protein[43] (Fig. 4, Supplementary Fig. 1). As confirmed for *No*LDSP, the clustering of small lipid droplets was independent of the presence or absence of the YFP-tag (Supplementary Fig. 2).

To compare different engineering approaches, the *A. grandis* abietadiene synthase was produced as plastid:*Ag*ABS (native), cytosol:*Ag*ABS$^{85-868}$ or LD:*Ag*ABS$^{85-868}$, each alone and combined with ER:*Ps*CYP720B4 (native), cytosol:*Ps*CYP720B4$^{30-483}$ or LD:*Ps*CYP720B4$^{30-483}$ + LD:*Ca*CPR$^{70-708}$ (Fig. 5). Note that these assays also included either *Pb*DXS + plastid:*Mt*GGDPS or *El*HMGR$^{159-582}$ + cytosol:*Mt*GGDPS to increase the precursor flux, and *At*WRI1$^{1-397}$ to initiate lipid droplet accumulation. *No*LDSP was included in those assays that lacked any *No*LDSP-fusion proteins. Compared to the assays with plastid:*Ag*ABS, production of cytosol:*Ag*ABS$^{85-868}$ and LD:*Ag*ABS$^{85-868}$ resulted in similar diterpenoid yield. When native or modified *A. grandis* abietadiene synthase was co-produced with native or modified *P. sitchensis* *Ps*CYP720B4, the leaves accumulated diterpene resin acids in free and glycosylated forms (Supplementary Figs. 3–5).

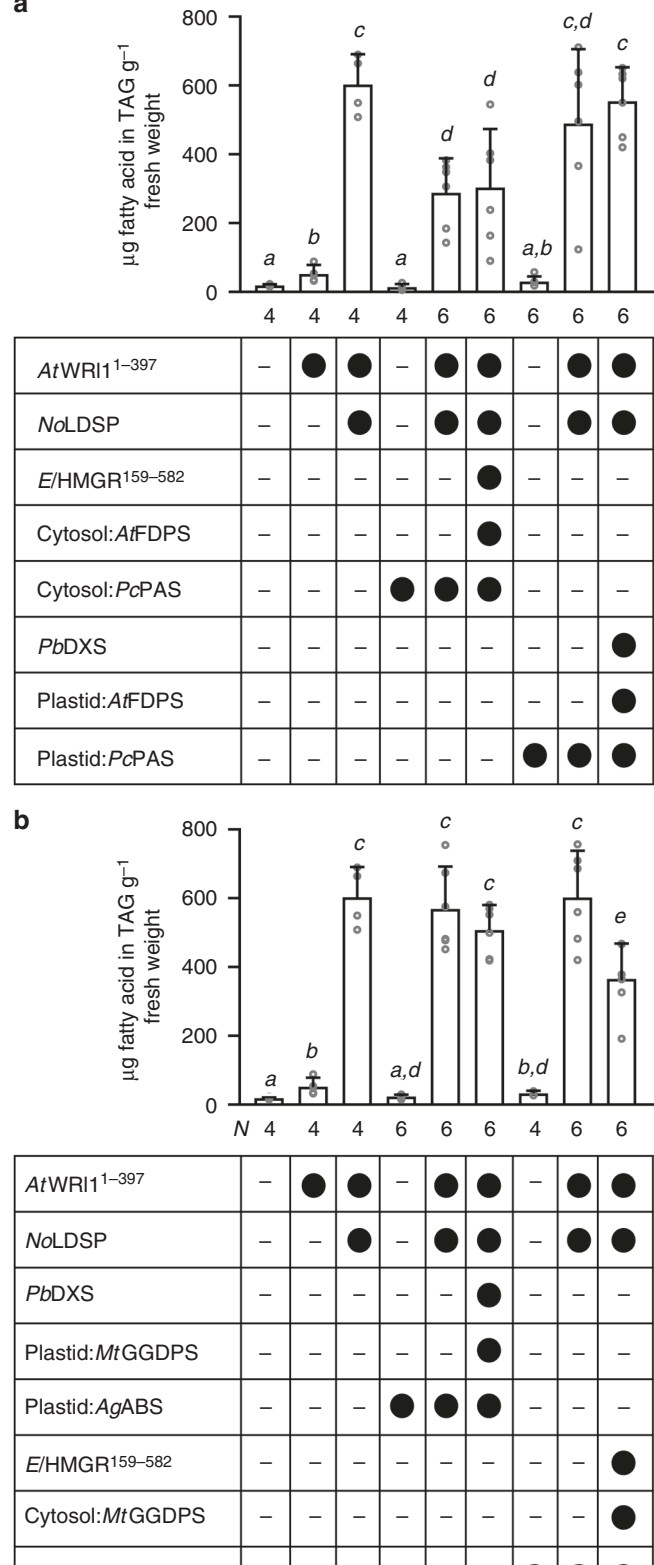

**Fig. 3** Triacylglycerol yield in engineered *N. benthamiana* leaves. TAG accumulation was initiated through ectopic expression of *WRINKLED1* (*AtWRI1*[1–397]) and further enhanced through co-expression of *No*LDSP. The impact of engineered patchoulol (**a**) and diterpenoid production (**b**) on TAG yield is depicted. The different construct combinations are indicated below each bar (black circle, was included; minus, was not included). Data were analyzed by Shapiro–Wilk, Welch–ANOVA (**a** $P < 0.0001$; **b** $P < 0.0001$) and Brown–Forsythe ANOVA (**a** $P < 0.0001$; **b** $P < 0.0001$) followed by *t*-tests (unpaired, two-tailed, Welch correction). Data are presented as individual biological replicates and bars representing average levels with SD (*N* indicated below each bar). This experiment was replicated twice. Statistically significant differences are indicated by *a-e* based on *t*-tests ($P < 0.05$). Source Data are provided as a Source Data file

acids which allowed determining the level of total diterpenoid acids produced in infiltrated leaves. To compare the different engineering strategies, the levels of both diterpenoids and total diterpenoid acids were quantified for each infiltrated leaf (Fig. 5). Co-production of plastid:*Ag*ABS with ER:*Ps*CYP720B4, cytosol:*Ps*CYP720B4[30–483] or LD:*Ps*CYP720B4[30–483] decreased the diterpenoid level (compared to controls with plastid:*Ag*ABS) and resulted in the accumulation of diterpenoid acids, consistent with diterpenoids being converted to diterpenoid acids. The level of diterpenoid acids was ~4-fold and 3-fold higher in transient assays with plastid:*Ag*ABS including ER:*Ps*CYP720B4 and plastid:*Ag*ABS + LD:*Ps*CYP720B4[30–483] + LD:*Ca*CPR[70–708] compared to assays including cytosol:*Ps*CYP720B4[30–483]. The highest diterpenoid acid yield in transient assays with cytosol:*Ag*ABS[85–868] was achieved in combination with ER:*Ps*CYP720B4 which was ~2- and 3-fold higher than with cytosol:*Ag*ABS[85–868] and LD:*Ps*CYP720B4[30–483] + LD:*Ca*CPR[70–708], respectively (Fig. 5). In transient assays with LD:*Ag*ABS[85–868], the diterpenoid acid level was 2-fold higher in assays with ER:*Ps*CYP720B4 than in assays with either cytosol:*Ps*CYP720B4[30–483] or LD:*Ps*CYP720B4[30–483] + LD:*Ca*CPR[70–708] (Fig. 5).

## Discussion

Our results demonstrate high-yield synthesis of target di- and sesquiterpenoids in engineered lipid droplet-accumulating leaves of *N. benthamiana* when precursor availability is enhanced. The flux of precursors into terpenoid synthesis was increased through co-production of de-regulated, robust enzymes from the MEP or MEV pathway (*Pb*DXS or *El*HMGR[159–582]) and GGDPS or FDPS in the same compartment. The data are consistent with previous studies in *N. benthamiana* reporting on the engineered production of diterpenoids (plastid-targeted) and a sesquiterpenoid (cytosol-targeted)[15,24,38]. Our comparative study with distinct GGDPSs indicates that a type I enzyme such as *Mt*GGDPS can be a robust alternative to type II GGDPS to increase precursor availability for diterpenoid synthesis and circumvent potential negative feedback (Fig. 2a, b). In principle, this approach can also be applied to optimize FDPS-dependent sesqui- or triterpenoid pathways.

Highest accumulation of the target sesquiterpenoid was achieved in this study through compartmentation of the biosynthetic pathway in the plastid instead of the cytosol (Fig. 1c). Diterpenoid production was targeted to the plastid (*Pb*DXS + plastid:*Mt*GGDPS + plastid:AgABS), cytosol or lipid droplets (*El*HMGR[159–582] + cytosol:*Mt*GGDPS + cytosol: *Ag*ABS[85–868]/LD:*Ag*ABS[85–868]) with similar success, yielding a high content of target diterpenoids in vegetative tissue. The anchoring of terpenoid biosynthesis enzymes on cytosolic lipid droplets in this study represents a promising approach in terpenoid biotechnology. It bears the potential to spatially re-arrange enzymes, bringing them

The glycosyl modifications of the diterpenoid acids were consistent with those previously reported for engineered terpenoid products and are likely the result of intrinsic defense/detoxification mechanisms in *N. benthamiana*[24,44,45]. Incubation of such leaf extracts with Viscozyme® L resulted in the hydrolysis of the glycosylated diterpenoid acids to free diterpenoid resin

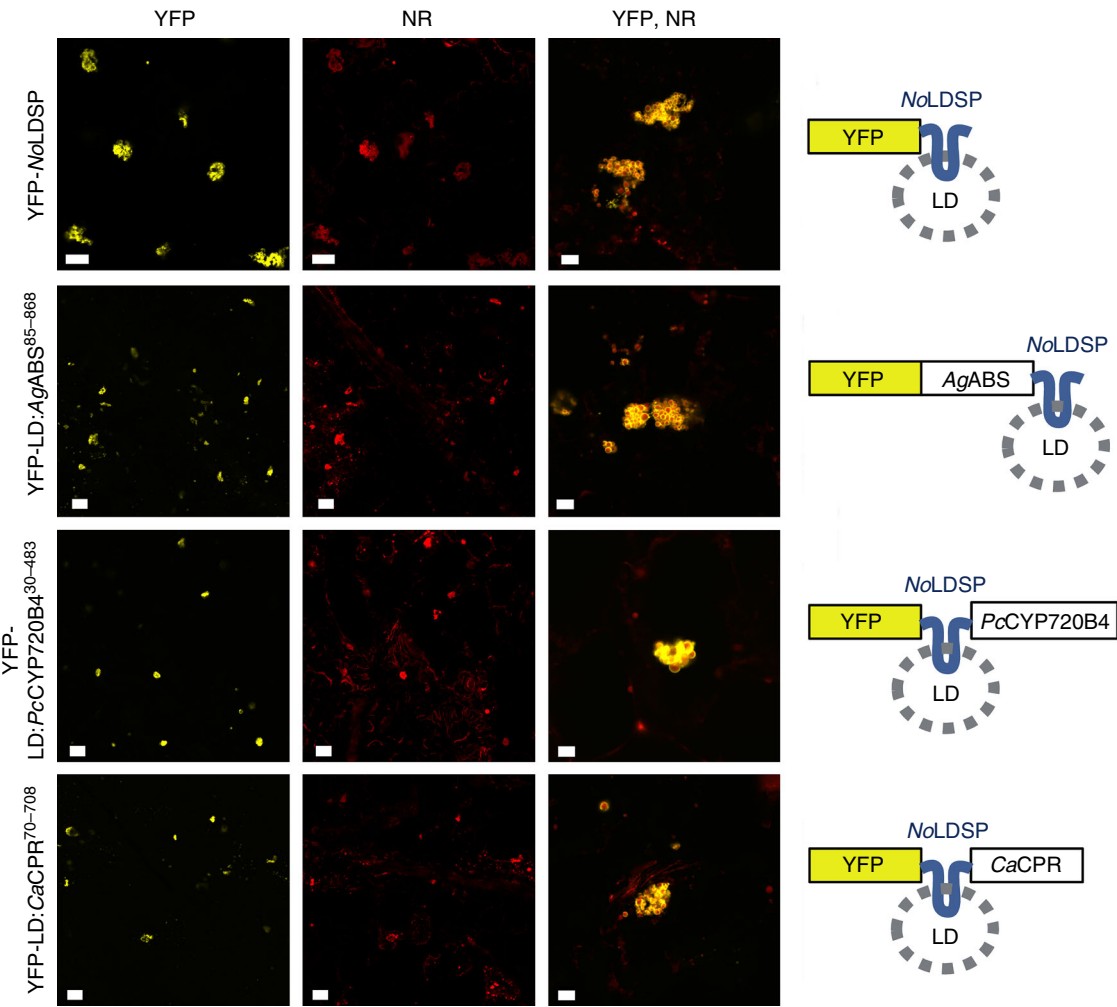

**Fig. 4** Localization of heterologously expressed fluorescent-reporter tagged fusion proteins. *N. benthamiana* leaves producing *yellow fluorescent protein* (YFP)-tagged *No*LDSP, LD:*Ag*ABS[85–868], LD:*Ps*CYP720B4[30–483], or LD:*Ca*CPR[70–708] were subjected to confocal laser scanning microscopy. Representative images are shown. The produced YFP-proteins are indicated in each line. Note that *At*WRI1[1–397] was co-produced and leaf samples were stained with Nile red to visualize neutral lipids in lipid droplets. This experiment was replicated twice. Channels: YFP yellow fluorescent protein (scale bar 20 µm), NR Nile red (scale bar 20 µm), YFP NR, enlarged merge YFP and NR (scale bar 5 µm)

into closer proximity and create multi-enzyme assemblies. The technology benefits from *No*LDSP's ability to (a) anchor fusion proteins on the lipid droplet surface, (b) stabilize small lipid droplets with relatively large surface-to-volume ratio, and (c) promote the clustering of small lipid droplets, thereby creating a large compartment-like structure (Fig. 4). Our results demonstrate that both diterpene synthase and CYP were catalytically active at lipid droplets (Fig. 5). Notably, the co-production of plastid:*Ag*ABS with either native *Ps*CYP720B4 (ER:*Ps*CYP720B4) or LD:*Ps*CYP720B4[30–483] resulted in similar diterpenoid acid yields. Targeting of the diterpene synthase to the cytosol or lipid-droplets impaired the production of diterpenoid acids at lipid droplets catalyzed by LD:*Ps*CYP720B4[30–483] (compared to ER:*Ps*CYP720B4). At this point, it remains unclear if the high-yield production of diterpenoids in the cytosol and their sequestration in the lipid droplets may have interfered with the functionality of the lipid-droplet targeted enzymes whereas plastid-targeting of the diterpene synthase may have resulted in a more favorable distribution of the diterpenoids between plastid, ER and lipid droplets under the selected experimental conditions. Overall, the top terpenoid yields in this study consolidate the versatility of the transient *N. benthamiana* system as a platform to produce terpenoids and test drive terpenoid biotechnology for later

production at industrial scales in economically relevant biomass crops. In this context, it must be noted that stable transgenic *N. benthamiana* engineered for plastid-targeted sesquiterpenoid production (plastid:FDPS + plastid:sesquiterpene synthase) exhibited shorter stature, chlorosis of the lower leaves and vein clearing, probably as a result of carbon competition between the engineered and essential native terpenoid pathways[15]. In high biomass crops, the use of inducible or weaker promoters (instead of strong constitutive promoters) in terpenoid engineering approaches may, therefore, help to prevent or reduce any interference with plant growth and development.

Co-engineering of lipid droplet synthesis in leaves influenced the target terpenoid yield to different extents depending on the applied engineering approach. Production of the diterpene synthase alone or together with GGDPS in the plastid or cytosol combined with high-yield lipid droplet synthesis (*At*WRI1[1–397] + *No*LDSP) enhanced the target diterpenoid yield up to 2.5-fold (Fig. 2b, c). Under these conditions, diterpenoids were sequestered in synthetic lipid droplets (Supplementary Fig. 1), which may have limited negative feedback and enhanced flux towards diterpenoid production. The abundance of engineered lipid droplets may potentially facilitate downstream processes to extract terpenoids from plant material through "trapping" of the target

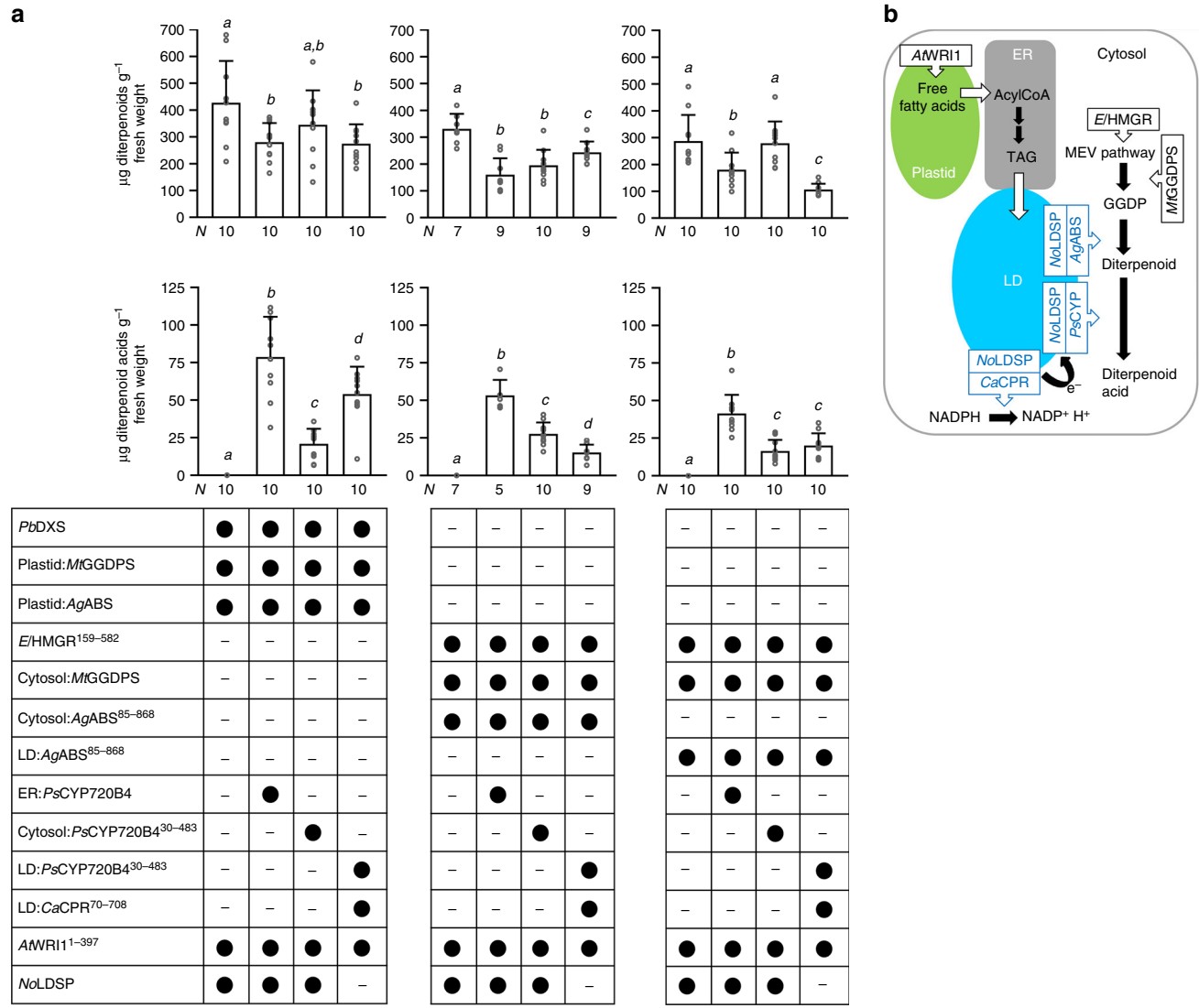

**Fig. 5** Lipid droplets as engineering platform for the production of functionalized diterpenoids. Terpenoid biosynthesis enzymes were produced as *No*LDSP-fusion proteins to target them to the lipid droplets (LD:*Ag*ABS[85–868], LD:*Ps*CYP720B4[430–483], and LD:*Ca*CPR[70–708]) and tested in different combinations as indicated below each bar (black circle, was included; minus, was not included) (**a**). Production of native or modified *Ag*ABS led to accumulation of diterpenoids and when native or modified *Ps*CYP720B4 was co-produced, to conversion of diterpenoids to diterpenoid acids. Data were analyzed by Shapiro–Wilk, Brown–Forsythe ANOVA (diterpenoids $P < 0.0184$, $P < 0.0001$, $P < 0.0001$; diterpenoid acids $P < 0.0001$, $P < 0.0001$, $P < 0.0001$) and Welch ANOVA (diterpenoids $P < 0.0509$, $P$ 0.0002, $P < 0.0001$; diterpenoid acids $P < 0.0001$, $P < 0.0001$, $P$ 0.0002) followed by $t$-tests (unpaired, two-tailed, Welch correction). Results are presented as individual biological replicates and bars representing average levels with SD ($N$ indicated below each bar). Statistically significant differences are indicated by $a$–$d$ based on $t$-tests ($P < 0.05$). This experiment was replicated twice. The scheme (**b**) depicts the conversion of abietadiene to abietic acid when LD:*Ag*ABS[85–868] (*No*LDSP-*Ag*ABS), LD:*Ps*CYP720B4[430–483] (*No*LDSP-*Ps*CYP) and LD:*Ca*CPR[70–708] (*No*LDSP-*Ca*CPR) were produced. Source Data are provided as a Source Data file. LD, lipid droplet; e⁻, electron from NADPH

compounds in the oil fraction. The findings are consistent with a recent publication reporting that co-engineering of sesquiterpenoid and lipid droplet production increased the yields of cytosol-derived volatile sesquiterpenoids by 2- to 4-fold[46]. In our study, production of the sesquiterpene synthase alone or together with FDPS combined with high-yield lipid droplet production did, however, not increase the sesquiterpenoid yield (Fig. 1b, c) suggesting that a certain ratio of sesquiterpenoids to lipid droplets may be critical to retain highly volatile sesquiterpenoids in the cytosol. Recently, an oleosin-based strategy in the plastid was successful to synthesize squalene, a triterpenoid, in plastids and to trap it in plastid lipid droplets[47]. The latter approach may be also suitable to trap plastid-derived volatile terpenoids in vegetative tissues.

When the plastid-targeted high-yield diterpenoid approach was combined with high-yield lipid droplet production, the diterpenoid yield significantly decreased and a trend towards a lower triacylglycerol level was determined (Figs. 2b and 3b). Co-engineering of the cytosol-targeted high-yield diterpenoid approach together with high-yield lipid droplet production did not affect the diterpenoid yield but resulted in a significantly lower triacylglycerol yield. The combination of the set of genes for high-yield sesquiterpenoid production with high-yield lipid droplet production negatively impacted the cytosol-derived sesquiterpenoid yield and resulted in an approximately 50% lower triacylglycerol yield (Figs. 1b and 3a) whereas the plastid-derived sesquiterpenoid yield was enhanced without triacylglycerol production being affected. In the plastids of engineered leaves, fatty

acid synthesis (initiated by AtWRI1[1−397]) and terpenoid synthesis likely competed directly for carbon (probably in form of pyruvate), resulting in an increase in the product yield from one pathway at the expense of the other pathway. Although both MEV pathway in the cytosol and fatty acid biosynthesis in the plastid require acetyl-CoA as precursor, a direct competition between these two pathways for acetyl-CoA appears implausible since acetyl-CoA is impermeable to membranes and independently produced and consumed in the different subcellular compartments and organelles of plant cells[48]. As metabolic interactions between the cytosol and plastid are not well understood, it remains unclear whether shifts in cytosolic metabolite pools may have influenced carbon partitioning between the cytosol and plastid.

## Methods

**Constructs for transient expression studies.** The open reading frames encoding truncated A. thaliana WRINKLED1 (AtWRI1[1−397], AY254038.2) and full-length N. oceanica lipid droplet surface protein (NoLDSP, JQ268559.1) were amplified from existing cDNAs[11,25]. The coding sequences for truncated cytosolic E. lathyris HMGR (ElHMGR[159−582], JQ694150.1), cytosolic A. thaliana FDPS (cytosol: AtFDPS, NM_117823.4), cytosolic P. cablin patchoulol synthase (cytosol:PcPAS, AY508730), plastidic A. grandis abietadiene synthase (plastid:AgABS, U50768.1), and plastidic P. barbatus (PbDXS) were amplified from cDNAs derived from total RNA of the host organisms. The cDNA sequence for PbDXS used in this study significantly differed from the previously published sequence[38] and was deposited in GenBank. The endoplasmic P. sitchensis CYP720B4 (ER:PsCYP720B4, HM245403.1) was amplified from a cDNA clone. The open reading frame encoding a truncated C. acuminata CPR (CaCPR[70−708], KP162177) lacking the N-terminal membrane anchor domain was synthesized. Codon optimized open reading frames were synthesized for the type I GGDPSs from S. acidocaldarius (SaGGDPS, D28748.1) and M. thermautotrophicus (MtGGDPS, AE000666.1) (Supplementary Data 1). A putative M. elongata AG77 MeGGDPS (type III) was identified through mining of transcriptome data[49] and a codon optimized open reading frame was synthesized (Supplementary Data 1). Two putative type II GGDPSs, EpGGDPS1 and EpGGDPS2, were identified through mining of E. peplus transcriptome data[50] and amplified from leaf cDNA. A putative type II GGDPS was identified in the genome of Tolypothrix sp. PCC 7601[51] and the coding sequence was amplified from genomic DNA. To target SaGGDPS, MtGGDPS, TsGGDPS, MeGGDPS, AtFDPS and PcPAS to the plastid, the sequences were fused at their N-terminus to the plastid targeting sequence of the A. thaliana ribulose bisphosphate carboxylase small chain 1A (NM_105379.4). The encoded plastid-targeted proteins are referred to as plastid:SaGGDPS, plastid:MtGGDPS, plastid:TsGGDPS plastid:MeGGDPS, plastid:AtFDPS and plastid:PcPAS. The coding sequences of A. grandis abietadiene synthase and P. sitchensis CYP720B4 (ER:PsCYP720B4) were truncated to target the enzymes to the cytosol, in this study referred to as cytosol:AgABS[85−868] and cytosol:PsCYP720B4[30−483], respectively. For lipid droplet targeting, truncated A. grandis abietadiene synthase, P. sitchensis CYP720B4 and C. acuminata CPR were either fused to the N-terminus or C-terminus of N. oceanica lipid droplet surface protein resulting in LD:AgABS[85−868], LD:PsCYP720B4[30−483] and LD:CaCPR[70−708], respectively (Fig. 4). All primers used in this study are described in Supplementary Table 1. The full-length and modified coding sequences were verified by sequencing, inserted into pENTR4 (Invitrogen), and subsequently transferred into the Gateway vectors pEarleygate 100 and pEarleygate 104 (N-terminal YFP-tag), each under control of a 35 S promoter for strong constitutive expression[52]. These constructs were introduced into A. tumefaciens LBA4404 for transient expression studies in N. benthamiana. Primers and constructs used in the study were designed with SnapGene 3.3.4.

**Transient expression in N. benthamiana leaves.** Transformants of A. tumefaciens LBA4404 carrying selected binary vectors were grown overnight at 28 °C in Luria-Bertani medium containing 50 μg/mL rifampicin and 50 μg/mL kanamycin. Prior to infiltration into N. benthamiana leaves, the A. tumefaciens cells were sedimented by centrifugation at 3800 × g for 10 min, washed, resuspended in infiltration buffer (10 mM MES-KOH pH 5.7, 10 mM MgCl₂, 200 μM acetosyringone) to an optical density at 600 nm (OD600) 0.8 and incubated for ~30 min at 30 °C. To test various gene combinations, equal volumes of the selected bacterial suspensions were mixed and infiltrated into N. benthamiana leaves using a syringe without a needle. A. tumefaciens LBA4404 carrying the tomato bushy stunt virus gene P19[53,54] was included in all infiltrations to suppress RNA silencing in N. benthamiana. The N. benthamiana plants used for infiltration were grown for 3.5 to 4 weeks in soil at 25 °C under a 12-h photoperiod at 150 μmol m⁻² s⁻¹. To compare different engineering strategies, only plants of the same batch were used in transient assays. Typically, three to five plants were used for each gene combination. To avoid developmental differences, the same two leaves were infiltrated on each plant. After infiltration, the plants were grown for 4 additional days in the growth chamber. Samples from the infiltrated leaves were subsequently analyzed

for terpenoid or triacylglycerol content. All experiments were conducted at least two times and the results are shown from representative experiments.

**Lipid analysis.** Triacylglycerol analyses were performed essentially as previously described with minor modifications[4]. For each sample, one infiltrated N. benthamiana leaf was freshly harvested and total lipids were extracted with 4 mL chloroform/methanol/formic acid (10:20:1, by volume). Ten microgram tri-17:0 triacylglycerol (Sigma) was added as internal standard to each sample. The total lipids were separated by thin layer chromatography on silica plates (Si250PA, Mallinckrodt Baker) developed with ether:ethyl ether:acetic acid (80:20:1, v/v/v). Triacylglycerol bands were visualized with a spraying dye (0.01% Primuline in 80% (v/v) acetone) under UV light. The TAG bands were scraped from the TLC plates and used to prepare fatty acid methyl esters by acid-catalyzed trans-methylation in 1 mL 1 M hydrochloric acid in anhydrous methanol at 80 °C for 25 min. The samples were extracted with 1 mL 0.9% sodium chloride and 1 mL hexane. After centrifugation at 1000 × g for 3 min, the hexane extract was collected, the volume was reduced under a stream of nitrogen and the extract was subjected to gas-liquid chromatography. Chromatography was performed with an Agilent DB-23 column at 48.6 mL min⁻¹ helium flow, 21.93 psi pressure and 250 °C inlet temperature. The following oven program was used: 2 min isothermal at 140 °C, 25 °C min⁻¹ to 160 °C, 8 °C min⁻¹ to 250 °C, 4 min isothermal at 250 °C followed by 38 °C min⁻¹ to 140 °C. The temperature of the flame ionization detector was 270 °C with 30.0 mL min⁻¹ hydrogen flow, 400 mL min⁻¹ air flow and 30.0 mL min⁻¹ helium flow. All triacylglycerol analysis was performed in Excel 2010.

**Statistical analyses.** Statistical analyses were conducted using Graphpad Prism 8 and included normality (Shapiro-Wilk), one-way ANOVA (Welch and Brown-Forsythe) and t-tests (unpaired, two-tailed, Welch correction). A P-value of < 0.05 was considered statistically significant.

**Terpenoid analyses in N. benthamiana leaves.** For each sample, 50 mg or 100 mg leaf tissue was incubated with 1 mL hexane containing 2 mg mL⁻¹ 1-eicosene (internal standard, TCI America) on a shaker for 15 min at room temperature prior to incubation in the dark for 16 h at room temperature. Sesquiterpenoids and diterpenoids were separated and analyzed by GC-MS using an Agilent 7890 A GC system coupled to an Agilent 5975 C MS detector. Chromatography was performed with an Agilent VF-5ms column (40 m × 0.25 mm × 0.25 μm) at 1.2 mL min⁻¹ helium flow. The injection volume was 1 μL in splitless mode at an injector temperature of 250 °C. The following oven program was used (run time 18.74 min): 1 min isothermal at 40 °C, 40 °C min⁻¹ to 180 °C, 2 min isothermal at 180 °C, 15 °C min⁻¹ to 300 °C, 1 min isothermal at 300 °C, 100 °C min⁻¹ to 325 °C, and 3 min isothermal at 325 °C. The mass spectrometer was operated at 70 eV electron ionization mode, a solvent delay of 3 min, ion source temperature at 230 °C, and quadrupole temperature at 150 °C. Mass spectra were recorded from m/z 30 to 600. Terpenoid products were identified based on retention times, mass spectra published in relevant literature and through comparison with the NIST Mass Spectral Library v17 (National Institute of Standards and Technology, USA). Quantitation of diterpenoid products and patchoulol was based on 1-eicosene standard curves. The extracted ion chromatograms for each target compound were integrated, and compounds were quantified using QuanLynx tool (Waters) with a mass window allowance of 0.2 and a signal-to-noise ratio ≥10. All calculated peak areas were normalized to the peak area for the internal standard 1-eicosene and tissue fresh weight.

Diterpenoid resin acids and glycosylated derivatives were analyzed by UHPLC/MS/MS to confirm accurate masses and fragments. For each sample, 100 mg leaf tissue and 1 mL methanol containing 1.25 μM telmisartan (internal standard, Toronto Research Chemicals) were added, mixed, and incubated in the dark at room temperature for 16 h. A 10-μL volume of each extract was subsequently analyzed using a 31-min gradient elution method on an Acquity BEH C18 UHPLC column (2.1 × 100 mm, 1.7 μm, Waters) with mobile phases consisting of 0.15% formic acid in water (solvent A) and acetonitrile (solvent B). The 31-min method gradient employed 1% B at 0.00 to 1 min, linear gradient to 99% B at 28.00 min, held until 30 min, followed by a return to 1% B and held from 30.10 to 31 min. The flow rate was 0.3 mL/min and the column temperature was 40 °C. The mass spectrometer (Xevo G2-XS QTOF, Waters) was equipped with an electrospray ionization source and operated in negative-ion mode. Source parameters were as follows: capillary voltage 2500 V, cone voltage 40 V, desolvation temperature 300 °C, source temperature 100 °C, cone gas flow 50 L/h, and desolvation gas flow 600 L/h. Mass spectrum acquisition was performed in negative ion mode over m/z 50 to 1500 with scan time of 0.2 s using a collision energy ramp 20 to 80 V. For quantitative analyses of the total diterpenoid resin acid level, 50 mg leaf tissue was incubated with 1 mL methanol/water (8/2, v/v) containing 2 μM telmisartan (internal standard, Toronto Research Chemicals) in the dark at room temperature. Note that a second sample was taken from each infiltrated leaf and subjected to diterpenoid analysis as described above. After 16 h, 200-μL aliquots of the methanol/water extracts were dried down under vacuum, reconstituted in 500 μL McIlvaine buffer (citrate phosphate buffer) pH 4.0 and incubated with 100 μL Viscozyme® L (Sigma Aldrich) at 37 °C for 16 h. Viscozyme® L is a multi-enzyme complex with a wide range of carbohydrase activities. After overnight incubation, samples were extracted with 500 μL dichloromethane, centrifuged for 10 min at

$4000 \times g$, 250-μL aliquots were then transferred to fresh glass vials, dried down under vacuum and resuspended in 50 μL 80% (v/v) methanol. A 10-μL volume of each extract was subsequently analyzed by UHPLC/MS/MS using a 16-min gradient elution method with mobile phases consisting of 10 mM ammonium formate in water (solvent A) and methanol (solvent B). The 16-min method gradient employed 20% B at 0.00 to 2 min, linear gradient to 99% B at 14.00 min, held until 15 min, followed by a return to 20% B and held from 15.10 to 16 min. The mass spectrometer was operated in negative-ion mode. MassLynx v4.1 was used for acquisition and processing of GC-MS and UHPLC/MS/MS data.

**Isolation of lipid droplets**. Lipid droplets were isolated as previously described[55] with minor adjustments[55]. For each sample, 1 g infiltrated *N. benthamiana* leaf tissue was ground with mortar and pestle in 20 mL ice-cold buffer A (20 mM tricine, 250 mM sucrose, 0.2 mM phenylmethylsulfonyl fluoride pH 7.8). The homogenate was filtered through Miracloth (Calbiochem) and centrifuged in a 50-mL tube at $3400 \times g$ for 10 min at 4 °C to remove cell debris. From each tube, 10 mL supernatant was collected and transferred to a 15-mL tube. The supernatant fraction was then overlaid with 3 mL buffer B (20 mM HEPES, 100 mM KCl, 2 mM $MgCl_2$, pH 7.4) and centrifuged for 1 h at $5000 \times g$. After centrifugation, 2 mL from the top of each gradient containing floating lipid droplets were collected. For terpenoid analysis, each lipid droplet fraction was extracted with 5 mL hexane containing 2 μg mL$^{-1}$ 1-eicosene (internal standard, TCI America) prior to GC-MS analysis. To avoid developmental differences, the same leaf from three different plants (biological replicates) were analyzed for each gene combination.

**Confocal imaging**. For lipid droplet visualization, sections of freshly harvested leaf samples were stained on microscope slides with Nile red solution (10 μg mL$^{-1}$ in phosphate buffered saline) in the dark. After one hour, the sections were briefly rinsed with phosphate buffered saline prior to microscopy. Imaging of Nile red, chlorophyll, and enhanced yellow fluorescent protein (EYFP) fluorescence was conducted with a confocal laser scanning microscope FluoView VF1000 (Olympus) at excitation 559 nm/emission 570–630 nm, excitation 559 nm/emission 655–755 nm, and excitation 515 nm/emission 527 nm, respectively. Images were processed using the FV10-ASW 4.2 microscopy software (Olympus).

**Reporting summary**. Further information on experimental design is available in the Nature Research Reporting Summary linked to this article.

## Data availability

Sequence data from this article were newly deposited in the GenBank/EMBL data libraries under the following accession numbers: MH363711 (*EpGGDPS1*), MH363712 (*EpGGDPS2*), MH363713 (*PbDXS*) and MH363714 (*TsGGDPS*). The codon optimized sequences for *SaGGDPS*, *MtGGDPS*, and *MeGGDPS* are given in Supplementary Data 1. The source data underlying Figs. 1, 2, 3 and 5 and Supplementary Fig. 1b are provided as a Source Data file.

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

## Acknowledgements

We thank Beronda Montgomery (Michigan State University) for the *Tolypothrix* sp. PCC 7601 cells, Jörg Bohlmann (University of British Columbia, Canada) for the *Ps*CYP720B4 cDNA, and Afton M. Dewyse, Robert Nolan, Jonathan A. Arnesen, Britta Hamberger, and Emily Lockwood at Michigan State University for technical assistance. We extend our thanks to Melinda Frame at the Center for Advanced Microscopy (Michigan State University) and A. Dan Jones and Tony Schilmiller at Michigan State University for assistance in the identification of the glycosylated diterpenoid acids. Mass spectra were collected using instruments at the Michigan State University Mass Spectrometry and Metabolomics core. This work was primarily supported by the U.S. Department of Energy-Great Lakes Bioenergy Research Center Cooperative Agreement DE-FC02-07ER64494 (B.H., C.B.) and partially by the Division of Chemical Sciences, Geosciences and Biosciences, Office of Basic Energy Sciences of the U.S. Department of Energy Grant DE- DE-FG02-91ER20021 (C.B.) and by AgBioResearch MICL02357 (C.B.), Michigan State University. B.H. gratefully acknowledges startup funding from the Department of Molecular Biology and Biochemistry, AgBioResearch (MICL02454) and the U.S. Department of Energy Grant (DE-SC0018409), Michigan State University.

## Author contributions

R.S. designed the study and performed the experiments, analyzed the data and wrote the article; R.S., P.K., J.C., and A.B. generated expression constructs, and Y.Y. performed triacylglycerol analysis. C.B. and B.H. conceived the study and edited the drafts of the article.

## Additional information

**Competing interests:** R.S., C.B., and B.H. are inventors on a provisional patent application at Michigan State University covering the findings discussed. The remaining authors declare no competing interests.

