## [Peer Review File · Nature Communications]

Reviewers' comments:

Reviewer #1 (Remarks to the Author):

The manuscript of Sadre et al. describes the development of several engineering strategies for the biotechnological production of terpenoids in plants. In general, terpenoid production in green plant tissues, such as leaves, is challenging primarily due to the dominance of photosynthetic biochemical, cellular and regulatory processes. The authors report the successful application of:

- (i) Enhanced oil droplet density using the WRINKLED regulator of plastid fatty acid metabolism and LDSP, a microalgal lipid droplet surface protein. Presumably, the enhanced formation and stability of the lipid droplets in tobacco leaves facilitated the increased accumulation of both sesquiterpenoids and diterpenoid produced either in plastids or in the cytosol.
- (ii) Enhanced availability of terpenoid precursors using selected variants, possessing desirable biochemical properties, of key enzymes operating in the mevalonate and non-mevalonate pathways, as appropriate.
- (iii) Targeting of diterpenoid biosynthetic enzymes to lipid droplets using LDSP fusion proteins, which facilitated the production of not only the terpenoid scaffold, but also functionalized diterpenoid acids produced using cytochromes P450 and the associated reductase partner.
- (iv) Various combinations of the above strategies.

Successful improvement and substantial increases in terpenoid accumulation for both volatile and non-volatile products was achieved. The authors provide a reasonable evaluation of ancillary aspects, such as an evaluation of the effect of increased terpenoid production of the triacylglycerol content of the leaves, and the cellular localization of LDSP fusion proteins to lipid droplets. Overall, the manuscript is clear, concise and well written. The senior author has considerable expertise in terpenoid metabolism, which certainly facilitated the selection of enzyme variants from a large collection of available parts, and the work is up to his usual high standards. The fact that two related papers have just been published by other groups shows the timeliness of the work and should be considered a factor in the efficient publication of the current work high-quality journal. The biotechnological directions presented in this manuscript will be valuable to a large number of other researchers working in the terpenoid field, and to efforts aimed at the commercial production of various terpenoids in heterologous plants. The more paradigmatic aspects (e.g. sequestering products in engineered organisms as a mechanism to reduced negative feedback bottlenecks) will be of broader interest. One important question, which I'm not sure the authors can answer adequately using a transient expression system, is the effect of their engineering efforts on the general growth and development of the host plant. Perhaps they could provide some commentary.

Reviewer #2 (Remarks to the Author):

The manuscript by Sadre et al. describes some interesting metabolic engineering approaches to optimize the biosynthesis of sesqui- and diterpenoid in *Nicotiana benthamiana* leaf tissue. The authors compare cytosolic vs plastidial engineered pathways and demonstrate that the latter results in increased substrate flux and higher terpenoid yields. In addition, the authors explore the effect of simultaneously boosting neutral lipid synthesis. Increased lipid synthesis causes trapping of terpenoids inside cytosolic lipid droplets, thereby acting as a cellular packaging vehicle. Finally, by reconstituting the diterpenoid and diterpenoid acid metabolic pathway on the lipid droplet surface, the authors claim increased diterpenoid conversion efficiency.

Though the paper contains some clever metabolic engineering designs, in my opinion these appear rather extensions of previously published attempts to engineer terpenoid biosynthesis in photosynthetic tissues. The authors point out in the discussion that several previous studies have already reported engineered cytosolic and/or plastidial pathways for the production of di- and sesquiterpenoids in *N. benthamiana* (references 16, 23, 38). In addition, the concept of trapping

terpenoids inside lipid droplets by co-expression of a lipid droplet protein has also been reported before (reference 40). Furthermore, this work does not contain any stable expression data. As such, the efficiency of the engineered pathways described in this paper remains to be proven in a stably transformed plant and any comparison with reported yields in stable transgenic lines (see p6 "... and approximately 7-fold higher than that achieved in stable transgenic *N. tabacum* lines¹⁶") should be interpreted with caution.

The use of lipid droplets as a scaffold for a multi-enzyme biosynthesis pathway such as terpenoid synthesis is an intriguing novel concept explored at the end of the manuscript. The authors conclude that an engineered pathway anchored onto the lipid droplet surface results in increased flux and conversion to diterpenoid acids. Though compelling, the evidence presented is indirect as the authors rely on reduced diterpenoid yields only to conclude increased conversion to diterpenoid acid. Based on the presented results, however, I don't think the authors can completely rule out the possibility of negative interactions occurring between AgABS, CYP720B4 and/or CPR when co-expressed. Impaired catalytic activity due to specific negative interactions could simply lead to lower levels of synthesized abietadiene, and would be wrongly interpreted as increased conversion to diterpenoid acid. Given the novelty of this concept, the authors should directly quantify abietic acid and its derivatives to support their claims.

Major comments:

1. Results p10 and Fig 4: "The YFP-signals in infiltrated leaves were subsequently compared to the signals obtained for YFP-tagged NoLDSP..." Rather than using chlorophyll fluorescence, the authors should stain leaf sections with a neutral lipid dye such as Nile Red to visualize lipid droplets and overlay with the YFP signal. This would confirm localization of the different LDSP fusion proteins to the surface of cytosolic neutral lipid droplets.
2. Results p11: "To facilitate the comparison between the different engineering strategies, the level of diterpenoids was determined instead of the diterpenoid acids (Figure 5)." The authors should quantify diterpenoid acid levels instead of diterpenoid levels for reasons outlined above.
3. Supplementary Figure 1b: What was the total LD yield for each of the treatments? How did the authors standardize the diterpenoid content found in each lipid droplet fraction to correct for possible differences in lipid droplet recovery efficiency during isolation? It would be better to report diterpenoid levels as ug/ug recovered neutral lipids for example.

Minor comments:

1. Introduction p2: "Plants accumulating high levels of terpenoids have evolved specialized anatomical features for their biosynthesis and storage including laticifer cells, resin ducts or cavities, and glandular trichomes". A reference would be helpful.
2. Results p6: "Leaves transiently producing PbDXS+plastid:AtFDPS+plastid:PcPAS+AtWRI11-397+NoLDSP yielded the highest patchouli level, an average 100-fold higher compared to leaves producing plastid:PcPAS and approximately 7-fold higher than that achieved in stable transgenic *N. tabacum* lines¹⁶". In the case of the plastidial engineered pathway, does patchouli also accumulate in plastoglobuli? Have the authors looked at the size and number of plastoglobuli in transiently transformed *N. benthamiana* mesophyll cells?
3. Results p7: "To enhance the production of diterpenoids, plastid:AgABS was co-produced in different combinations with PbDXS and a plastid GGDPS." The schematic drawing in Fig2a mentions CfDXS.
4. Results p7: "...five predicted GGDPSs from the archaea *Methanothermobacter thermoautotrophicus*..." Only one *M. thermoautotrophicus* GGDPS is mentioned in Fig2a.
5. Results p8: "Inclusion of PbDXS and plastid:MtGGDPS in the transient assays yielded the highest diterpenoid level independent of whether lipid droplets were co-engineered (Figure 2b)." It does not appear to be 'independent' of whether or not lipid droplets were co-engineered since coexpression of WRI1 and LDSP resulted in a significant reduction of the leaf diterpenoid content
6. Results p9: "In plastid-targeted approaches for high-yield sesquiterpenoid and diterpenoid synthesis, a less pronounced trend towards lower triacylglycerol yield was determined (Figures 3a and 3b)." This is not the case for sesquiterpenoid synthesis. Coexpression of WRI1 and LDSP with

either cytosolic or plastidial engineered patchoulol biosynthetic pathway resulted in a similar TAG decrease (Fig 3a).

7. Results p10: "The NoLDSP-fusion proteins are here referred to as LD:AgABS85-868, LD:PsCYP720B430-483 and LD:CaCPR70-708." Can the authors explain why they used N- vs C-terminal fusions in these specific instances?

8. Results p10: "..., consistent with a previous report on ectopic production of *A. thaliana* OLEOSIN1 fused to green fluorescent protein¹² (Figure 4, Supplemental Figures 1 and 2)." should read "... (Figure 4, Supplemental Figures 2 and 3).

9. Discussion p12: "In addition, the abundance of engineered lipid droplets can potentially facilitate downstream processes to efficiently extract terpenoids from plant material through "trapping" of the target compounds in the oil fraction (Supplemental Figure 1)." Cost-efficient extraction and downstream processing would require high 'trapping efficiencies' into neutral lipid droplets. The authors should determine the amount of terpenoids 'trapped' in the lipid droplets as a ratio of the total leaf terpenoid content.

10. Discussion p12: "The results of transient assays indicated that the engineered synthesis of lipid droplets had no significant impact on high-yield terpenoid synthesis in the plastid (Figures 1c and 2b)." The last treatment in Fig 2b rather suggests a significant reduction compared to the third treatment in the same experiment.

11. Discussion p12: "In contrast to the plastid-targeted approach, cytosolic sesquiterpenoid synthesis (but not diterpenoid synthesis) was strongly impaired when lipid droplet accumulation was co-engineered..." Can the authors speculate as to why they observed this difference?

12. Methods p16: "Samples from the infiltrated leaves were subsequently analyzed for terpenoid or triacylglycerol content." What is the typical experimental outline when testing different gene combinations in *N. benthamiana*? Do the different treatments originate from different leaves and/or plants or are all treatments infiltrated in the same leaf to avoid leaf/plant developmental differences?

13. Supplemental Fig 4: "...producing AtWRI11-397 + EIHMGR + plastid:MtGGDPS + LD:AgABS + ER:PsCYP720B4." This particular gene combination does not correspond to any of the combinations tested in Fig 5.

Reviewer #3 (Remarks to the Author):

The study by Sadre and colleagues attempts to show that mechanisms that trigger the biosynthesis and accumulation of lipids in lipid droplets can be used for the efficient production and accumulation of terpenoids in the leaves of plants. The study promotes this approach as a breakthrough for the generation of biomass crops that accumulate industrially relevant terpenoids. There are several issues raised in this review about the evidence provided to support this hypothesis. In addition, there may be better approaches already available in Nature that have led to high levels of diterpenoid production in plants, such as those found for *Stevia rebaudiana* that accumulates 10 to 30 % of their leaf dry weight as steviolosides.

1. Figure 1 Please indicate if the transient expression experiments represent biological replicates or technical replicates. I expect these are biological replicates?

2. Figure 1a This experiment shows that co-expression of AtWRI1 and NoLDSP stimulates a 12-fold increase in TAG accumulation compared to controls.

- What results would be expected if only NoLDSP were expressed? This is important to establish to show that both genes need to be expressed to obtain the described results.
- The experiment should include data detecting or attempting to detect the production of these proteins in tobacco leaves.

3. Figure 1b and c reports how the transient expression of different genes in the cytosol vs the plastid affect the production of patchoulol. The basic finding is that patchoulol accumulation increases incrementally when lipid droplets are also produced. The authors assume that their

engineering strategies did sequester target terpenoids to lipid droplets, but provide no evidence that patchoulol is not emitted as shown in the previous ground-breaking study (Ref 16). Nevertheless, the data is not convincing that lipid droplets can successfully sequester this class of volatile metabolite to a much higher degree (Fig 1c, lane 9) than in their absence (Fig 1c, lane 3). In this context the previous report (Ref 16) describes transgenic plants that appear to be quite active in patchoulol biosynthesis, but much of this product appears to be emitted. It seems that a simple solution for accumulating patchoulol in transgenic tobacco would be to engineer the required glucosyltransferase that would diminish product volatility and increase its sequestration in plant vacuoles? Finally, it is probably not appropriate to compare transient production of this compound versus its stable production in transformed plants. Production of stable transgenic tobacco lines expressing 5 separate genes would be much more difficult to achieve, but may be necessary to make such comparisons.

4. Figure 2: Please indicate if the transient expression experiments represent biological replicates or technical replicates. I expect these are biological replicates?

2. Figure 2a demonstrates the importance of transient co-expression of DXS and GGPS genes for the biosynthesis and accumulation of diterpenes. The data is used to formulate transient expression to see the effect of oil droplet production on their accumulation.

3. Figures 2b and 2c both indicate that the presence of oil droplets (Fig 2b, lanes 8 and 9 and Fig 2c lanes 8 and 9) do not enhance diterpene accumulation compared to the control experiment (Fig 2b, lane 3 and Fig 2c lane 3). It is difficult to reconcile these results with the interpretations provided in the results section, with claims that diterpenoid levels are higher with the presence of lipid drops, even if diterpene levels have risen 2-fold in leaves producing plastid:AgABS+AtWRI11-397+NoLDSP compared to control leaves producing plastid:AgABS. The appropriate control to compare to should be Fig 2b, lane 3 and Fig 2c lane 3.

4. The statement "Inclusion of PbDXS and plastid:MtGGDPS in the transient assays yielded the highest diterpenoid level independent of whether lipid droplets were co-engineered (Figure 2b)." suggests that lipid droplets do not assist in increasing diterpenoid production.

Modify the following:

1. "Due to the economical relevance as high-density biofuel, strategies have been established to enhance the accumulation of triacylglycerol in vegetative tissues of high-biomass yielding crops³⁻⁵." To "Due to the possible economic relevance lipids for use as a high-density biofuel, strategies have been established to enhance the accumulation of triacylglycerol in vegetative tissues of high-biomass yielding crops³⁻⁵."

2. "A primary target for engineering was WRINKLED1" to A primary target for increasing lipid production and accumulation was genetic engineering WRINKLED1 expression,

3. "Yields of triacylglycerol were further increased by removal of an intrinsically disordered region of Arabidopsis thaliana WRINKLED1 (AtWRI11-397) increasing the protein's stability¹¹ and coproduction with ectopic lipid biosynthesis enzymes and a lipid droplet associated protein^{3,9,12}." "Yields of triacylglycerol were further increased by removal of an intrinsically disordered region of Arabidopsis thaliana WRINKLED1 (AtWRI11-397) shown to increase the protein's stability¹¹ and that enhances expression of ectopic lipid biosynthesis enzymes and a lipid droplet associated protein^{3,9,12}."

Reviewer #4 (Remarks to the Author):

The work presented in the article entitled "Cytosolic lipid droplets as engineered organelles for production and accumulation of terpenoid biomaterials in plant leaves" demonstrate an exciting new way to trap volatile molecules in vegetative tissue. The demonstration is performed with agro-infiltration of various combinations of enzymes in the leaves of *Nicotiana Benthamiana*. Among the numerous novelties showcased in this work the functional fusion of a truncated P450 on a lipid droplet is particularly exciting.

Yet some small point would need addressing:

Major points:

- I was wondering why the authors did not use a DGAT1 enzyme to further boost the production of TAGs in the *N. benthamiana* system? Based on the available literature (some of it cited in this paper) the combination would be more potent.
- In the figure 2a heterologous lipid production is not induced with NoLDSP/AtWRI1 the presence of a blue LD in miss leading.
- Lines 239-240 the authors assume that the decrease in diterpenoids is due to a direct conversion to resin acids. This is clearly to facilitate the readers understanding. Yet, here the reader would like to dispel doubts. The direct link between decreasing diterpenoids and increasing resin acids needs to be established without the show of doubt. This would best done in a quantitative manner for at least the key combinations.

Minor points:

- CfDXS is present in the figures and the legends but in the manuscript and table indicating which constructs were used one can read PdDXS. Please correct.
- Line 655 page 32 there is no + in the figure but a black circle
- Figure 5. It would be interesting to observe if the presence of NoLDS with the protein fusion has a beneficial effect on the accumulation of products.
- Is the presence of LD:CPR influencing the conversion of diterpenoids?
It would have been interesting to test this.
- Line 271 based on the results presented here and the likelihood that a very large part of the produced molecule are emitted in the atmosphere this assumption need to be amended.
- Both the MVA (MEV) and TGA biosynthetic pathway use Acyl-CoA as a precursor
- I am missing the accession number for the PbDXS

REVIEWER 1:

One important question, which I'm not sure the authors can answer adequately using a transient expression system, is the effect of their engineering efforts on the general growth and development of the host plant. Perhaps they could provide some commentary.

A commentary is included in the discussion section of the revised manuscript to address the reviewer's comment. Wu et al. (2006) reported that stable transgenic Nicotiana benthamiana engineered for plastid-targeted sesquiterpenoid production (plastid:FDPS+plastid:sesquiterpene synthase) exhibited shorter stature, leaf chlorosis and vein clearing. The described phenotype was probably caused by carbon competition between the engineered and essential native terpenoid pathways.

REVIEWER 2 (Remarks to the Author):

Major comments:

- (1) Results p10 and Fig 4: "The YFP-signals in infiltrated leaves were subsequently compared to the signals obtained for YFP-tagged NoLDSP..." Rather than using chlorophyll fluorescence, the authors should stain leaf sections with a neutral lipid dye such as Nile Red to visualize lipid droplets and overlay with the YFP signal. This would confirm localization of the different LDSP fusion proteins to the surface of cytosolic neutral lipid droplets.**

The new Figure 4 in the revised manuscript shows the localization of the YFP-tagged LDSP-fusion proteins on the surface of Nile red stained lipid droplets.

- (2) Results p11: "To facilitate the comparison between the different engineering strategies, the level of diterpenoids was determined instead of the diterpenoid acids (Figure 5)." The authors should quantify diterpenoid acid levels instead of diterpenoid levels for reasons outlined above.**

We agree with reviewer #2. Figure 5 shows now both the level of diterpenoids and total diterpenoid acids in infiltrated leaves. The main text and materials and methods section have been modified to describe the new experiments.

- (3) Supplementary Figure 1b: What was the total LD yield for each of the treatments? How did the authors standardize the diterpenoid content found in each lipid droplet fraction to correct for possible differences in lipid droplet recovery efficiency during isolation? It would be better to report diterpenoid levels as ug/ug recovered neutral lipids for example.**

The revised material and methods section and the figure legend describe in more detail the experimental outline for the isolation of lipid droplets and quantification of terpenoids in the lipid droplet fraction. In supplementary Figure 1b, each bar represents the average terpenoid content (mean±SD) in the lipid droplet fraction isolated from one gram leaf material from three biological replicates. Each biological replicate was collected from an independent plant, same leaf.

Due to the lack of engineered lipid droplets in the control, a comparison of diterpene accumulation normalized to recovered lipids was not informative. We feel more comfortable expressing the terpenoid level as $\mu\text{g terpenoid g}^{-1}$ fresh weight throughout the manuscript. To provide context, terpenoid and TAG yield in lipid droplet producing samples and controls are shown in Figures 1, 2 and 3, respectively.

Minor comments:

- 1. Introduction p2: “Plants accumulating high levels of terpenoids have evolved specialized anatomical features for their biosynthesis and storage including laticifer cells, resin ducts or cavities, and glandular trichomes”. A reference would be helpful.**

*A reference has been included in the revised manuscript: Lange, B.M. The evolution of plant secretory structures and emergence of terpenoid chemical diversity. *Annu. Rev. Plant Biol.* **66**, 139-159 (2015).*

- 2. Results p6: “Leaves transiently producing PbDXS+plastid:AtFDPS+plastid:PcPAS+AtWRI11-397+NoLDSP yielded the highest patchoulol level, an average 100-fold higher compared to leaves producing plastid:PcPAS and approximately 7-fold higher than that achieved in stable transgenic *N. tabacum* lines16”. In the case of the plastidial engineered pathway, does patchoulol also accumulate in plastoglobuli? Have the authors looked at the size and number of plastoglobuli in transiently transformed *N. benthamiana* mesophyll cells?**

As the lipids in chloroplast plastoglobuli consist of triacylglycerol and various prenyl lipids (van Wijk et al. 2018), it is likely that patchoulol would also accumulate in plastoglobuli. We have not investigated the size and number of plastoglobuli but refer the reader in the discussion section to a relevant publication by Zhao et al. (2018) who engineered lipid droplets in plastids to trap triterpenoids.

*van Wijk KJ, Kessler F. Plastoglobuli: Plastid Microcompartments with Integrated Functions in Metabolism, Plastid Developmental Transitions, and Environmental Adaptation. *Annu Rev Plant Biol.* 2017 Apr 28;68:253-289*

- 3. Results p7: “To enhance the production of diterpenoids, plastid:AgABS was co-produced in different combinations with PbDXS and a plastid GGDPS.” The schematic drawing in Fig2a mentions CfDXS.**

CfDXS was replaced by PbDXS in the schematic drawing of Figure 2a.

- 4. Results p7: “...five predicted GGDPSs from the archaea *Methanothermobacter thermoautotrophicus*...” Only one *M. thermoautotrophicus* GGDPS is mentioned in Fig2a.**

The text has been modified to describe in more detail the origin of the selected GGDPs.

5. Results p8: “Inclusion of PbDXS and plastid:MtGGDPS in the transient assays yielded the highest diterpenoid level independent of whether lipid droplets were co-engineered (Figure 2b).” It does not appear to be ‘independent’ of whether or not lipid droplets were co-engineered since coexpression of WRI1 and LDSP resulted in a significant reduction of the leaf diterpenoid content.

The text was modified for clarity. Under the conditions for high-yield production of diterpenoids, inclusion of AtWRI1¹⁻³⁹⁷ had no negative impact on diterpenoid accumulation whereas inclusion of AtWRI1¹⁻³⁹⁷+NoLDSP resulted in a significant reduction of the diterpenoid level (relative to leaves containing PbDXS+ plastid:MtGGDPS+ plastid:AgABS).

6. Results p9: “In plastid-targeted approaches for high-yield sesquiterpenoid and diterpenoid synthesis, a less pronounced trend towards lower triacylglycerol yield was determined (Figures 3a and 3b).” This is not the case for sesquiterpenoid synthesis. Coexpression of WRI1 and LDSP with either cytosolic or plastidial engineered patchouli biosynthetic pathway resulted in a similar TAG decrease (Fig 3a).

The text was revised to address the reviewer’s comment and Figure 3b was corrected. The approaches for sesquiterpenoid synthesis in the cytosol combined with the production of WRI1 and LDSP resulted in decreased TAG levels. A combination of the approaches for plastid-targeted high-yield sesquiterpenoid and lipid droplet production had, however, no significant impact on TAG yield compared to controls.

In leaves producing WRI1 and LDSP, diterpenoid production in the plastid and low-yield production of diterpenoids in the cytosol had no significant impact on TAG yield. High-yield synthesis of diterpenoids in the cytosol resulted in a significant lower TAG yield.

7. Results p10: “The NoLDSP-fusion proteins are here referred to as LD:AgABS⁸⁵⁻⁸⁶⁸, LD:PsCYP720B4³⁰⁻⁴⁸³ and LD:CaCPR⁷⁰⁻⁷⁰⁸.” Can the authors explain why they used N- vs C-terminal fusions in these specific instances?

*The manuscript explains now in more detail the design of the NoLDSP-fusion proteins. Previous studies provide evidence that N-terminal or C-terminal tagged terpene synthases were functional. For example, it was demonstrated that a C-terminal tag did not compromise the activity of a bifunctional diterpene synthase (Gnanasekaran et al. 2015) that shares 85% identity on amino acid sequence level with the bifunctional *Abies grandis* abietadiene synthase (AgABS) used in the present study. On the basis of this finding, LD:AgABS⁸⁵⁻⁸⁶⁸ was designed as C-terminal NoLDSP-fusion protein. To re-target PsCYP720B4 to lipid droplets, the predicted N-terminal hydrophobic domain of native PsCYP720B4 was replaced by NoLDSP. The design of the NoLDSP-fusion protein LD:PsCYP720B4³⁰⁻⁴⁸³ was inspired by Gnanasekaran et al. 2015 who demonstrated that modifying or deleting the membrane anchoring domain of PsCYP720B4 did not impair the enzyme’s activity in planta. In our study, the predicted N-terminal hydrophobic domain of native CaCPR was replaced by NoLDSP to re-target CaCPR to lipid droplets. Data reported by Qu et al.*

(2015) on N-terminal and C-terminal tagged CaCPR (after removal of a predicted N-terminal hydrophobic domain) indicated that the C-terminus of CaCPR is pivotal for reductase activity and not suitable for modifications.

8. Results p10: “..., consistent with a previous report on ectopic production of *A. thaliana* OLEOSIN1 fused to green fluorescent protein¹² (Figure 4, Supplemental Figures 1 and 2).” should read “... (Figure 4, Supplemental Figures 2 and 3).

“(Figure 4, Supplemental Figures 1 and 2)” was replaced by (Figure 4, Supplemental Figure 1) in the revised manuscript. Note that a new Figure 4 was prepared that made the former Supplemental Figure 2 obsolete.

9. Discussion p12: “In addition, the abundance of engineered lipid droplets can potentially facilitate downstream processes to efficiently extract terpenoids from plant material through “trapping” of the target compounds in the oil fraction (Supplemental Figure 1).” Cost-efficient extraction and downstream processing would require high ‘trapping efficiencies’ into neutral lipid droplets. The authors should determine the amount of terpenoids ‘trapped’ in the lipid droplets as a ratio of the total leaf terpenoid content.

The analysis of isolated lipid droplet fractions from biological replicates indicated a significant enrichment of target terpenoids in the oil layer (Supplemental Figure 1). It is therefore reasonable to speculate that the abundance of engineered lipid droplets may potentially facilitate downstream processes to extract terpenoids from plant materials through “trapping” of the target compounds in the oil fraction. Cost-efficient extraction and downstream processing will depend on various other factors than “trapping efficiencies” into lipid droplets, for example on the efficiency of mechanical or chemical disruption of the plant material.

We agree with the reviewer that our initial statement may have been misleading and modified the sentence in the revised manuscript to “The abundance of engineered lipid droplets may potentially facilitate downstream processes to extract terpenoids from plant material through “trapping” of the target compounds in the oil fraction. “

10. Discussion p12: “The results of transient assays indicated that the engineered synthesis of lipid droplets had no significant impact on high-yield terpenoid synthesis in the plastid (Figures 1c and 2b).” The last treatment in Fig 2b rather suggests a significant reduction compared to the third treatment in the same experiment.

The manuscript has been revised to provide a more accurate description and discussion of the results.

11. Discussion p12: “In contrast to the plastid-targeted approach, cytosolic sesquiterpenoid synthesis (but not diterpenoid synthesis) was strongly impaired when lipid droplet accumulation was co-engineered...” Can the authors speculate as to why they observed this difference?

The level and extent of metabolic interactions between the plastid and cytosol are not well understood. In a recent publication, Henry et al. (2018) reported that plants regulate the ratio of IDP to IP (and probably also the ratios of DMADP to DMAP and FDP to FP based on in vitro data). It is possible that shifts in the metabolite pools may have influenced carbon partitioning in different ways.

12. Methods p16: “Samples from the infiltrated leaves were subsequently analyzed for terpenoid or triacylglycerol content.” What is the typical experimental outline when testing different gene combinations in *N. benthamiana*? Do the different treatments originate from different leaves and/or plants or are all treatments infiltrated in the same leaf to avoid leaf/plant developmental differences?

The text was modified to describe in more detail the experimental outline. For each gene combination, three to five plants were used. To avoid developmental differences, the same two leaves were infiltrated on each plant and later analyzed. A total of six to ten leaves (from 3 to 5 plants) were analyzed for each gene combination as indicated in the figure legends.

13. Supplemental Fig 4: “...producing AtWRI11-397 + ElHMGR + plastid:MtGGDPS + LD:AgABS + ER:PcCYP720B4.” This particular gene combination does not correspond to any of the combinations tested in Fig 5.

The description of the gene combination used in the experiment was corrected (now Supplemental Figure 3).

REVIEWER 3:

1. Figure 1 Please indicate if the transient expression experiments represent biological replicates or technical replicates. I expect these are biological replicates?

The figure legend was modified to address the reviewer’s comment. Average levels are shown with SD for biological replicates.

2. Figure 1a This experiment shows that co-expression of AtWRI1 and NoLDSP stimulates a 12-fold increase in TAG accumulation compared to controls.

- **What results would be expected if only NoLDSP were expressed? This is important to establish to show that both genes need to be expressed to obtain the described results.**
- **The experiment should include data detecting or attempting to detect the production of these proteins in tobacco leaves.**

The role of WRINKLED1 and its effect on TAG accumulation in engineered plants has been extensively studied in various plant species including Nicotiana benthamiana. In particular, Ma et al. (2015) reported the localization of AtWRI1¹⁻³⁹⁷ and the increase in TAG production in transiently engineered Nicotiana benthamiana leaves. Our data are consistent with the previous findings. The localization of YFP-tagged LDSP on the periphery of lipid droplets is shown in Figure 4 and consistent with a previous publication reporting on the localization of LDSP in stably transformed Arabidopsis thaliana (Vieler et al. 2012). The present manuscript focuses primarily on the use of NoLDSP as a tool to anchor biosynthetic steps on lipid droplets. In this context, we confirmed that production of NoLDSP would not have any negative impact on AtWRI1¹⁻³⁹⁷-initiated TAG accumulation (Figure 1a).

1. Figure 1 Please indicate if the transient expression experiments represent biological replicates or technical replicates. I expect these are biological replicates?

The figure legend was modified to include this information. Average levels are shown with SD for biological replicates.

3. Figure 1b and c reports how the transient expression of different genes in the cytosol vs the plastid affect the production of patchouliol. The basic finding is that patchouliol accumulation increases incrementally when lipid droplets are also produced. The authors assume that their engineering strategies did sequester target terpenoids to lipid droplets, but provide no evidence that patchouliol is not emitted as shown in the previous ground-breaking study (Ref 16). Nevertheless, the data is not convincing that lipid droplets can successfully sequester this class of volatile metabolite to a much higher degree (Fig 1c, lane 9) than in their absence (Fig 1c, lane 3). In this context the previous report (Ref 16) describes transgenic plants that appear to be quite active in patchouliol biosynthesis, but much of this product appears to be emitted. It seems that a simple solution for accumulating patchouliol in transgenic tobacco would be to engineer the required glucosyltransferase that would diminish product volatility and increase its sequestration in plant vacuoles? Finally, it is probably not appropriate to compare transient production of this compound versus its stable production in transformed plants. Production of stable transgenic tobacco lines expressing 5 separate genes would be much more difficult to achieve, but may be necessary to make such comparisons.

Co-engineering of plastid-targeted patchouliol production and lipid droplet production improved the patchouliol yield (Figure 1c). A very recent publication by Delatte et al. (2018) shows that co-engineering of sesquiterpenoid and lipid droplet production increased the yields of cytosol-derived volatile sesquiterpenoids several fold. In our study, cytosol-targeted sesquiterpenoid approaches combined with lipid droplet production resulted in significantly lower sesquiterpenoid and TAG yield which suggests that a certain ratio of sesquiterpenoids to lipid droplets may be critical to retain highly volatile sesquiterpenoids in the cytosol.

The glycosylation of terpenoid products would cost additional carbon and thus, increase the metabolic burden in engineered plant cells.

We followed the reviewer's advice and modified the text to omit a comparison of data from transient assays with those published for transgenic tobacco lines.

4. Figure 2: Please indicate if the transient expression experiments represent biological replicates or technical replicates. I expect these are biological replicates?

The figure legend was modified to include this information. Average levels are shown with SD for biological replicates.

5. Figure 2a demonstrates the importance of transient co-expression of DXS and GGPS genes for the biosynthesis and accumulation of diterpenes. The data is used to formulate transient expression to see the effect of oil droplet production on their accumulation.

Figures 2b and 2c both indicate that the presence of oil droplets (Fig 2b, lanes 8 and 9 and Fig 2c lanes 8 and 9) do not enhance diterpene accumulation compared to the control experiment (Fig 2b, lane 3 and Fig 2c lane 3). It is difficult to reconcile these results with the interpretations provided in the results section, with claims that diterpenoid levels are higher with the presence of lipid drops, even if diterpene levels have risen 2-fold in leaves producing plastid:AgABS+AtWRI11-397+NoLDSP compared to control leaves producing plastid:AgABS. The appropriate control to compare to should be Fig 2b, lane 3 and Fig 2c lane 3.

The statement "Inclusion of PbDXS and plastid:MtGGDPS in the transient assays yielded the highest diterpenoid level independent of whether lipid droplets were co-engineered (Figure 2b)." suggests that lipid droplets do not assist in increasing diterpenoid production.

The text was modified for clarity. Co-production of AtWRI11-397+NoLDSP together with plastid:AgABS or plastid:MtGGDPS+ plastid:AgABS increased diterpenoid production up to 2.5-fold (compared to assays without AtWRI11-397+NoLDSP), consistent with lipid droplets promoting diterpenoid accumulation (sequestration). This beneficial effect of lipid droplet production on diterpenoid production was not achieved when PbDXS was included. Co-production of PbDXS+plastid:MtGGDPS+plastid:AgABS in combination with AtWRI11-397+NoLDSP impaired diterpenoid production (compared to PbDXS+plastid:MtGGDPS+plastid:AgABS) and resulted in a decreasing trend in TAG synthesis suggesting that under these conditions, the production of diterpenoids and TAG compete for carbon.

Modify the following:

1. "Due to the economical relevance as high-density biofuel, strategies have been established to enhance the accumulation of triacylglycerol in vegetative tissues of high-biomass yielding crops³⁻⁵." To "Due to the possible economic relevance lipids for use as a high-density biofuel, strategies have been established to enhance the accumulation of triacylglycerol in vegetative tissues of high-biomass yielding crops³⁻⁵."

Following the reviewer's suggestion, the sentence was modified.

2. "A primary target for engineering was WRINKLED1" to A primary target for increasing lipid production and accumulation was genetic engineering WRINKLED1 expression,

The sentence was modified according to the reviewer's suggestion.

3. "Yields of triacylglycerol were further increased by removal of an intrinsically disordered region of Arabidopsis thaliana WRINKLED1 (AtWRI11-397) increasing the protein's stability¹¹ and coproduction with ectopic lipid biosynthesis enzymes and a lipid droplet associated protein^{3,9,12}." "Yields of triacylglycerol were further increased by removal of an intrinsically disordered region of Arabidopsis thaliana WRINKLED1 (AtWRI11-397) shown to increase the protein's stability¹¹ and that enhances expression of ectopic lipid biosynthesis enzymes and a lipid droplet associated protein^{3,9,12}."

The text was modified for clarity.

REVIEWER 4 (Remarks to the Author):

1. I was wondering why the authors did not use a DGAT1 enzyme to further boost the production of TAGs in the N. benthamiana system? Based on the available literature (some of it cited in this paper) the combination would be more potent.

The role and effect of ectopic expression of WRINKLED1 on TAG accumulation has been extensively studied in various plant species (including Nicotiana benthamiana) in the past decade. WRINKLED1 was demonstrated to efficiently initiate lipid droplet accumulation. We aimed for a minimum number of genes for TAG production as our study primarily focused on the design of engineering approaches for terpenoid production. The data of our study show that under certain conditions, terpenoid and TAG production compete for carbon, an aspect that has to be considered in future applications.

2. In the figure 2a heterologous lipid production is not induced with NoLDSP/AtWRI1 the presence of a bleu LD is misleading.

Figure 2a was modified and the bleu LD was removed.

3. Lines 239-240 the authors assume that the decrease in diterpenoids is due to a direct conversion to resin acids. This is clearly to facilitate the readers understanding. Yet, here the reader would like to dispel doubts. The direct link between decreasing diterpenoids and increasing resin acids needs to be established without the show of doubt.

We agree with reviewer #4 and prepared a new Figure 5 to show both the level of diterpenoids and total diterpenoid acids in infiltrated leaves. The main text and materials and methods section in the revised manuscript were modified and describe the new results and experimental setup.

Minor points:

- **CfDXS is present in the figures and the legends but in the manuscript and table indicating which constructs were used one can read PdDXS. Please correct.**

*CfDXS was replaced by PbDXS (*Plectranthus barbatus* DXS) throughout the revised manuscript.*

- **Line 655 page 32 there is no + in the figure but a black circle**

The legend of Figure 3 was corrected to replace + by a black circle.

- **Figure 5. It would be interesting to observe if the presence of NoLDS with the protein fusion has a beneficial effect on the accumulation of products.**

Our results indicated that the NoLDSP-fusion proteins were correctly targeted to the lipid droplets (Figure 4 and Figure 5). Moreover, the resulting lipid droplets (and lipid droplet clusters) were in their appearance indistinguishable from those with NoLDSP (Figure 4). It seems therefore unlikely that NoLDSP would have an effect distinct from that of the NoLDSP-fusion proteins.

- **Is the presence of LD:CPR influencing the conversion of diterpenoids? It would have been interesting to test this.**

Our results are consistent with recent literature demonstrating that cytochromes P450 from plants require the activity of co-localized CPRs for productivity in non-native hosts or synthetic compartments (Renault et al. 2014, Bavishi et al. 2016).

- **Line 271 based on the results presented here and the likelihood that a very large part of the produced molecule are emitted in the atmosphere this assumption needs to be amended.**

The discussion section has been extensively revised and the original line 271 has been amended.

- **Both the MVA (MEV) and TGA biosynthetic pathway use Acyl-CoA as a precursor.**

This point is discussed in the revised manuscript. Cytosol and plastid independently produce and consume acetyl-CoA (Oliver et al. 2009). A direct competition between mevalonate pathway in the cytosol and fatty acid biosynthesis in the plastid is implausible as acetyl-CoA is membrane-impermeable.

- **I am missing the accession number for the PbDXS**

The accession number was included but given as CfDXS accession number. In the revised manuscript, CfDXS has been replaced by PbDXS throughout the manuscript.

REVIEWERS' COMMENTS:

Reviewer #2 (Remarks to the Author):

The authors have addressed all my questions

Reviewer #3 (Remarks to the Author):

The manuscript has been edited and modified to respond to my previous review comments.

Reviewer #4 (Remarks to the Author):

The diligent work of the authors yielded a manuscript which- in my view -is well suited for the current journal choice. TL Delatte